# Functional characterization of neuropeptides that act as ligands for both calcitonin-type and pigment-dispersing factor-type receptors in a deuterostome

Xiao Cong[1,2†], Huachen Liu[1†], Lihua Liu[1], Nayeli Escudero Castelán[3], Kite GE Jones[3], Michaela Egertová[3], Maurice R Elphick[3]*, Muyan Chen[1]*

[1]The Key Laboratory of Mariculture, Ministry of Education, Ocean University of China, Qingdao, China; [2]Yantai Institute of Coastal Zone Research, Chinese Academy of Sciences, Beijing, China; [3]School of Biological and Behavioural Sciences, Queen Mary University of London, London, United Kingdom

*For correspondence:
m.r.elphick@qmul.ac.uk (MRE);
chenmuyan@ouc.edu.cn (MC)

†These authors contributed equally to this work

Competing interest: The authors declare that no competing interests exist.

## eLife Assessment

This **valuable** study characterises receptors for calcitonin-related peptides from a deuterostomian animal, the echinoderm *Apostichopus japonicus*, by a combination of heterologous expression, pharmacological experiments, and the quantification of gene-expression levels. The authors provide **convincing** evidence for a functional calcitonin-related peptide system in the sea cucumber, but further work will be needed to confirm the proposed physiological functions of PDF receptor system in this species. This work should be of interest to scientists studying the signaling pathways, functions, and evolution of neuropeptides, and could be of relevance to improving the culture conditions of this economically key species.

**Abstract** The calcitonin (CT) family of related peptides exerts diverse physiological effects in mammals via two G-protein-coupled receptors: CTR and the CTR-like receptor CLR. Phylogenetic analysis of CT-type signaling has revealed the presence of CT-type peptides and CTR/CLR-type proteins in both deuterostome and protostome invertebrates. Furthermore, experimental studies have demonstrated that in the protostome *Drosophila melanogaster,* the CT-like peptide DH$_{31}$ can act as a ligand for a CTR/CLR-type receptor and a pigment-dispersing factor (PDF) receptor. Here, we investigated the signaling mechanisms and functions of CT-type neuropeptides in a deuterostome invertebrate, the sea cucumber *Apostichopus japonicus* (phylum Echinodermata). In *A. japonicus*, a single gene encodes two CT-type peptides (AjCT1 and AjCT2), and both peptides act as ligands for a CTR/CLR-type receptor (AjCTR) and two PDF-type receptors (AjPDFR1, AjPDFR2), but with differential activation of downstream cAMP/PKA, Gαq/Ca$^{2+}$/PKC, and ERK1/2 signaling pathways. AjCT1/AjCT2-encoding transcripts were detected in the central nervous system and a variety of organ systems, and neuropeptide expression was visualized immunohistochemically using an antiserum to a starfish CT-type peptide (ArCT). In vitro pharmacological experiments demonstrated that AjCT1 and/or AjCT2 cause dose-dependent relaxation of longitudinal muscle and intestine preparations. Furthermore, in vivo pharmacological experiments, combined with gain- and loss-of-function experiments, revealed a potential physiological role for AjCT2/AjPDFR2 signaling in promoting feeding and growth in *A. japonicus*. To our knowledge, this is the first study to obtain evidence that

CT-type peptides can act as ligands for both CTR/CLR-type and PDF-type receptors in a deuterostome. Moreover, it provides the first evidence for appetite-stimulating and growth-promoting effects of CT-type neuropeptides in bilaterians. Given the economic importance of *A. japonicus* as a foodstuff, the discovery of CT-type peptides as potential regulators of feeding and growth in this species may offer novel strategies for aquaculture applications.

## Introduction

The peptide calcitonin (CT) was first identified as a hormone that regulates blood calcium levels and subsequently a family of CT-related neuropeptides was discovered in mammals, including calcitonin gene-related peptides (CGRPs), amylin (AMY), adrenomedullin (AM), and CT receptor-stimulating peptide (CRSP; *Copp et al., 1962*; *Hay et al., 2004*; *Sekiguchi et al., 2009*; *Sekiguchi et al., 2016*; *Sekiguchi, 2022*). Members of the CT-type family of peptides typically have a disulfide bond in their N-terminal region (*Cai et al., 2018*) and exert their physiological effects in mammals by binding to two related G protein-coupled receptors: CTR (CT receptor) and CLR (CTR-like receptor; *Martins et al., 2014*). However, receptor activity-modifying proteins (RAMPs) acting as accessory proteins are key determinants of ligand selectivity for CTR and CLR (*McLatchie et al., 1998*). Thus, formation of a complex with a RAMP is essential for CLR function, whereas RAMPs are not required for CTR ligand recognition (*Kuwasako et al., 2011*; *Bomberger et al., 2012*). Furthermore, binding of CT to CTR in the absence of RAMPs (*Katafuchi et al., 2009*) can activate signaling via several downstream pathways, including cAMP accumulation, $Ca^{2+}$ mobilization, and ERK activation (*Chen et al., 1998*; *Pondel, 2000*; *Walker et al., 2010*; *Andreassen et al., 2014*).

Transcriptome/genome sequencing has enabled analysis of the phylogenetic distribution of CT-type signaling systems, indicating that the evolutionary origin of CT-type peptides and CTR/CLR-related receptors can be traced back to the urbilaterian common ancestor of deuterostomes and protostomes (*Furuya et al., 2000*; *Johnson et al., 2005*; *Jékely, 2013*; *Mirabeau and Joly, 2013*). Furthermore, in protostomes, two different types of CT-related peptides have been identified. Firstly, peptides that are structurally similar to vertebrate CT-type peptides in having an N-terminal disulphide bridge and secondly, CT-like diuretic hormone 31 ($DH_{31}$)-type peptides that lack an N-terminal disulphide bridge (*Furuya et al., 2000*; *Conzelmann et al., 2013*). Furthermore, phylogenetic analysis indicates that gene duplication in a common ancestor of the protostomes gave rise to genes encoding CT-type peptides with a disulphide bridge and $DH_{31}$-type peptides, which were then differentially retained in protostome taxa (*Conzelmann et al., 2013*; *Cai et al., 2018*). Receptors for CT-type peptides with a disulphide bridge have been characterized in a molluscan species, the oyster *Crassostrea gigas* (now known as *Magallana gigas*), which has seven CTR/CLR-type receptors. Two of these receptors, named Cg-CT-R and Cragi-CTR2, were shown to be specifically activated by the CT-type neuropeptides Cragi-CT1b and Cragi-CT2, respectively, thereby mediating intracellular $Ca^{2+}$ mobilization and cAMP accumulation (*Schwartz et al., 2019*). CT-type peptides with a disulphide bridge are not present in the insect *Drosophila melanogaster*. However, receptors that are activated by $DH_{31}$ have been identified in this species and, interestingly, $DH_{31}$ can act as a ligand for two types of receptors. Thus, $DH_{31}$ acts as a ligand for a CTR/CLR-related protein (CG17415), but it also acts as a ligand for a protein (CG13758) that is the receptor for pigment-dispersing factor (PDF; *Johnson et al., 2005*; *Mertens et al., 2005*; *Shafer et al., 2008*). Furthermore, the functional significance of the existence of dual receptor signaling pathways for $DH_{31}$ in *Drosophila* has been investigated extensively (*Johnson et al., 2005*; *Goda et al., 2019*).

Currently, relatively little is known about CT-type signaling systems in deuterostome invertebrates. Analysis of transcriptome/genome sequence data has enabled identification of genes/transcripts encoding CT-type peptides and candidate CTR/CLR-type receptors for these peptides in several deuterostome invertebrate taxa, including urochordates (e.g. *Ciona intestinalis*), cephalochordates (e.g. *Branchiostoma floridae*), hemichordates (e.g. *Saccoglossus kowalevskii*), and echinoderms (e.g. *Strongylocentrotus purpuratus*; *Burke et al., 2006*; *Sekiguchi et al., 2009*; *Sekiguchi et al., 2016*; *Rowe and Elphick, 2012*; *Martins et al., 2014*). Furthermore, analysis of the expression and action of CT-type peptides identified in the starfish *Asterias rubens*, *Acanthaster planci* and *Patiria pectinifera* has revealed that they act as muscle relaxants (*Cai et al., 2018*; *Semmens et al., 2016*). However, experimental confirmation of ligand-receptor partners has thus far only been reported for

the cephalochordate *B. floridae*, where three CT-type peptides were shown to act as ligands for a CTR/CLR-type receptor when it is co-expressed with RAMP-type proteins (*Sekiguchi et al., 2016*). To gain further insights into the mechanisms of CT-type signaling in invertebrate deuterostomes, the aim of this study was to functionally characterize CT-type peptides and their receptors in an echinoderm – the sea cucumber *Apostichopus japonicus*.

*A. japonicus* is an economically important species in Asian aquaculture (*Pangestuti and Arifin, 2018*; *Guo et al., 2020*) and a species representative of the phylum Echinodermata. As a phylum of deuterostome invertebrates, echinoderms form a sister clade to the chordates together with hemichordates and xenoturbellids (*Bromham and Degnan, 1999*; *Bourlat et al., 2006*; *Dunn et al., 2008*) and as such they occupy a key 'intermediate' phylogenetic position relative to the protostome invertebrates and vertebrates (*Zandawala et al., 2017*), which make them attractive as experimental systems to address evolutionary questions. In our previous studies, we have established this species as a model system to obtain insights into the evolutionary history and comparative physiology of neuropeptide signaling systems in bilaterians (*Chen et al., 2019*; *Li et al., 2022*; *Zheng et al., 2022*; *Cong et al., 2023*; *Zheng et al., 2024*). A gene encoding two CT-type peptides (AjCT1 and AjCT2) has been identified in *A. japonicus* (*Chen et al., 2019*), and a candidate receptor (AjCTR) for these peptides has also been predicted (*Huang et al., 2021*). Furthermore, relevant to the discovery that the CT-like peptide DH$_{31}$ can act as a ligand for both a CTR/CLR-type receptor and a PDF-type receptor in *Drosophila*, the sequences of two PDF-type receptors have been identified in *A. japonicus* (AjPDFR1 and AjPDFR2) (*Huang et al., 2021*).

The first aim of this study was to investigate if AjCT1 and/or AjCT2 act as ligands for AjCTR, AjPDFR1, and AjPDFR2 and to characterize the downstream signaling pathways that are activated when AjCT1 and/or AjCT2 bind to their receptor(s). The second aim was to investigate the physiological roles of AjCT1 and AjCT2 in *A. japonicus* by analyzing the expression of the gene encoding these peptides, investigating CT-type neuropeptide expression using immunohistochemistry, examining the in vitro and in vivo pharmacological effects of these peptides, and performing loss-of-function tests. The functional characterization of CT-type signaling in *A. japonicus* reported in this study provides new insights into the evolution and comparative physiology of neuropeptides and a basis for application of neuropeptide-based strategies to improve aquaculture methods for this economically important species.

## Results

### Characterization and phylogenetic analysis of AjCT1/2

The complete cDNA sequences of the CT-type neuropeptide precursors *AjCTP1* and *AjCTP2* were confirmed. The coding sequence of *AjCTP2* is shorter than *AjCTP1* (*Figure 1A*) but both transcripts encode precursor proteins that comprise the same predicted 23-residue N-terminal signal peptide (*Figure 1—figure supplement 1A*). *AjCTP1* encodes both putative CT-type neuropeptides (AjCT1 and AjCT2), while *AjCTP2* only encodes AjCT2 (*Figure 1A*, *Figure 1—figure supplement 1A and B*). Exon and intron structure analysis showed that *AjCTP1* comprises four exons, with AjCT1 and AjCT2 encoded in the third and fourth exons, respectively. *AjCTP2* comprises three exons, excluding the third exon that encodes AjCT1 (*Figure 1A*). Phylogenetic analysis revealed that AjCTP1 and AjCTP2 are closely related to CT-type precursors from other sea cucumbers (*Holothuria glaberrima* and *Holothuria scabra*) and are positioned within a clade that also contains CT-type precursors from other echinoderms (sea urchin *S. purpuratus*, starfish *A. rubens*). Furthermore, the echinoderm CT-type precursors are positioned within a clade that also comprises CT-type precursors from chordates, whilst DH$_{31}$/CT-type precursors from protostomes are positioned in a separate clade (*Figure 1B*). Sequence alignment of CT-related neuropeptides revealed that, with the exception of DH$_{31}$-type peptides, the N-terminal region of CT-related peptides has a pair of conserved cysteine residues that are known or predicted to form a disulfide bond. Another conserved feature is a C-terminal proline residue, with the exception of CGRPs (*Figure 1—figure supplement 2*). Accordingly, both AjCT1 and AjCT2 have these conserved characteristics.

### Characterization and phylogenetic analysis of three candidate receptors for AjCT1/2

The sequences of three candidate receptors for AjCT1 and AjCT2 were obtained by RACE and phosphorylation sites were predicted, as shown in *Figure 2—figure supplement 1A, B and C*. Phylogenetic

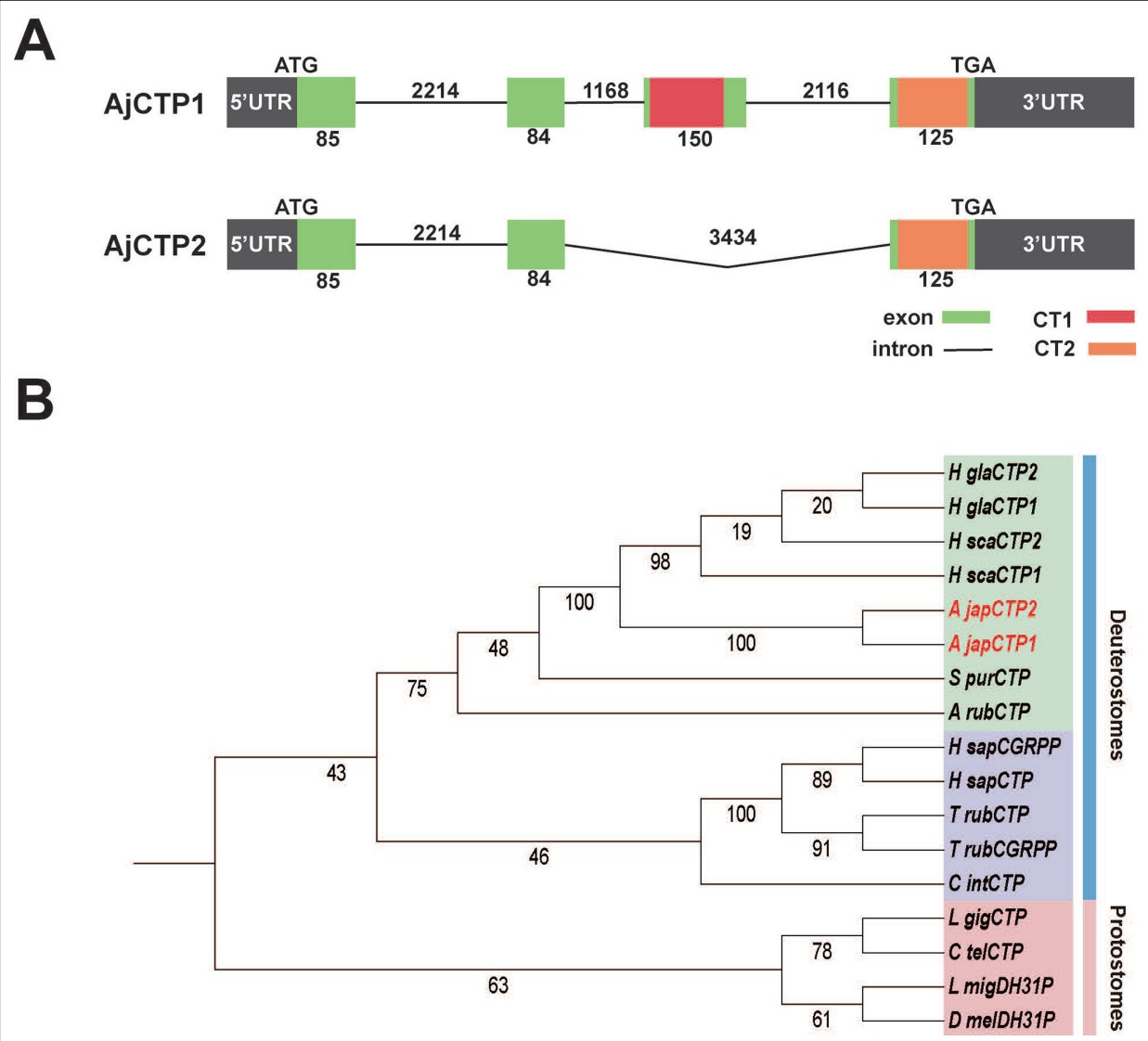

**Figure 1.** CT-type neuropeptide precursors and candidate receptors for CT-type peptides in *A. japonicus*. (**A**) Alternative splicing schematic of *AjCTP1* and *AjCTP2*. Exons are represented by green rectangles, introns are represented by lines, AjCT1 and AjCT2 are represented by red and orange rectangles respectively, upstream and downstream non-coding regions are represented by gray rectangles. The numbers represent the length of exons or introns. (**B**) Phylogenetic analysis of AjCTP1 and AjCTP2 and CT-related neuropeptide precursors from other bilaterians. CT-type precursors from deuterostomes are shown in green (echinoderms) and purple (chordates), CT-type precursors from protostomes are shown in pink. Full species names and accession numbers are listed in *Figure 1—source data 1*.

The online version of this article includes the following source data and figure supplement(s) for figure 1:

**Source data 1.** Accession numbers or citations of calcitonin-type family peptides or precursors in Bilateria.

**Figure supplement 1.** Calcitonin-type (CT-type) neuropeptide precursors in *A. japonicus*.

**Figure supplement 2.** Comparison of the sequences CT-type neuropeptides from Bilateria: deuterostomes-ambulacraria (green), deuterostomes-chordates (purple), and protostomes (pink).

analysis revealed that CTR/CLR-related receptors divided into three clades and an *A. japonicus* CTR/CLR-type receptor (AjCTR) was positioned in a clade with other deuterostome CTR/CLR-type receptors, including human CTR and CLR. The other two clades comprised protostome DH31-type receptors, which include the *Drosophila* receptor that has been identified experimentally as the receptor for DH31, and other protostome CTR/CLR-related receptors (Cluster A), including two receptors in the mollusc *Magallana gigas* that have been identified experimentally as receptors for CT-like peptides with a disulphide bridge. The phylogenetic analysis also revealed that two *A. japonicus* PDF-type receptors

(AjPDFR1 and AjPDFR2) were positioned in a clade comprising PDF-type receptors, with one branch containing PDF-type receptors from deuterostomes (including AjPDFR1 and AjPDFR2) and another branch containing PDF-type receptors from protostomes (including the *Drosophila* PDF receptor; *Figure 2*). Domain prediction of the candidate receptors showed that AjCTR had HRM (Hormone receptor) and 7tm_GPCRs (seven-transmembrane G-protein-coupled receptor) superfamily domains (*Figure 2—figure supplement 2A*), whereas AjPDFR1 and AjPDFR2 only presented the 7tm_GPCRs superfamily domain (*Figure 2—figure supplement 2B*). To further investigate an orthologous relationship of the candidate receptors with homologous receptors in other echinoderms, we conducted a gene synteny analysis in *A. japonicus*, *A. rubens,* and *L. variegatus*. The neighboring genes of *AjCTR* included *GOLGA4*, *NeuroD*, *CRF2*, *ppGalNAc-T*, *URB3*, *SLC40A1*, *CMAH,* and *MAGP*. The relative positions of these genes were different from those in *A. rubens* and *L. variegatus* (*Figure 2—figure supplement 2C*). The same situation was also observed for the synteny analysis of PDFRs.

## AjCT1 and AjCT2 act as ligands for AjCTR, AjPDFR1, and AjPDFR2

To investigate if AjCT1 and/or AjCT2 act as ligands for AjCTR, AjPDFR1, and AjPDFR2, here, two receptor assay methods were employed. As shown in *Figure 3A*, both AjCT1 and AjCT2 ($10^{-6}$ M) caused a significant increase in CRE luciferase activity in HEK293T cells transfected with AjCTR, AjPDFR1, or AjPDFR2. Furthermore, this effect was inhibited by pre-incubation of cells with the PKA inhibitor H89. The intracellular accumulation of cAMP was also measured, and this showed that AjCT1 and AjCT2 ($10^{-9}$ – $10^{-5}$ M) caused a dose-dependent increase in cAMP in HEK293T cells transfected with AjCTR, AjPDFR1, or AjPDFR2 (*Figure 3B*). However, when exposed to AjCT1 or AjCT2, the maximum accumulation of intracellular cAMP was higher in cells transfected with AjCTR than in cells transfected with AjPDFR1 and AjPDFR2 (*Figure 3B*). Measurement of SRE luciferase activity revealed that AjCT1 and AjCT2 ($10^{-6}$ M) administration evoked an increase in activity in HEK293T cells transfected with AjCTR and AjPDFR1 (*Figure 4A*). However, in HEK293T cells transfected with AjPDFR2, an increase in SRE luciferase activity was only observed with AjCT2, not with AjCT1. Furthermore, as shown in *Figure 4A*, AjCT1- and AjCT2-induced increases in SRE luciferase activity were not observed in cells that had been pre-incubated with FR900359 (Gαq protein inhibitor) and Gö 6983 (PKC inhibitor). As shown in *Figure 4—figure supplement 1*, receptors localized in the cell membrane, but after exposure to AjCT1 and AjCT2 ($10^{-6}$ M) for 15 min, internalization of pEGFP-N1/*AjCTR*, pEGFP-N1/*AjPDFR1,* and pEGFP-N1/*AjPDFR2* was observed in transfected HEK293T cells (*Figure 4B*). Taken together, all these findings indicate that AjCT1 and AjCT2 can act as ligands for the three candidate receptors and then activate cAMP/PKA signaling and Gαq/Ca$^{2+}$/PKC signaling in all combinations, with the exception of AjCT1/AjPDFR2 which only activates the cAMP/PKA cascade.

To investigate whether the activated signaling pathways may exert downstream effects via the ERK–MAPK pathway, the phosphorylation levels of ERK1/2 were analyzed in transfected HEK293T cells. pERK1/2 levels in cells transfected with AjCTR or AjPDFR1, but not AjPDFR2, were significantly increased after a 5 min incubation with AjCT1 (*Figure 5A*). Furthermore, this effect was blocked by the PKC inhibitor in AjCTR-expressing cells, and this effect was blocked by both PKA and PKC inhibitors in AjPDFR1-expressing cells (*Figure 5—figure supplement 1A*). pERK1/2 levels in cells transfected with AjCTR or AjPDFR1 or AjPDFR2 were significantly increased after a 5 min incubation with AjCT2 (*Figure 5B*). Moreover, this effect persisted until 15 min after exposure to AjCT in both AjCTR-transfected cells and AjPDFR1-transfected cells but not in AjPDFR2-transfected cells (*Figure 5B*, *Table 1*). Furthermore, these effects were blocked by PKA and PKC inhibitors in AjPDFR1 and AjPDFR2 expressing cells, whereas in AjCTR expressing cells, peptide-induced pERK1/2 activity was only blocked by a PKC inhibitor (*Figure 5—figure supplement 1B*). These results indicate that, with the exception of AjCT1/AjPDFR2, both AjCT1 and AjCT2 can activate the ERK1/2 cascade by binding to the candidate receptors.

## Expression profile of AjCTP1/2 and its receptors in *A. japonicus*

qRT-PCR analysis revealed that *AjCTP1/2* had the highest expression level in the circumoral nervous system (CNS; *Figure 6A*). The expression levels of the three receptors differed from those of *AjCTP1/2*. *AjCTR* (*Figure 6B*) and *AjPDFR1* (*Figure 6C*) showed the highest expression level in CNS and longitudinal muscle, whereas *AjPDFR2* exhibited the highest expression in the respiratory tree (*Figure 6D*).

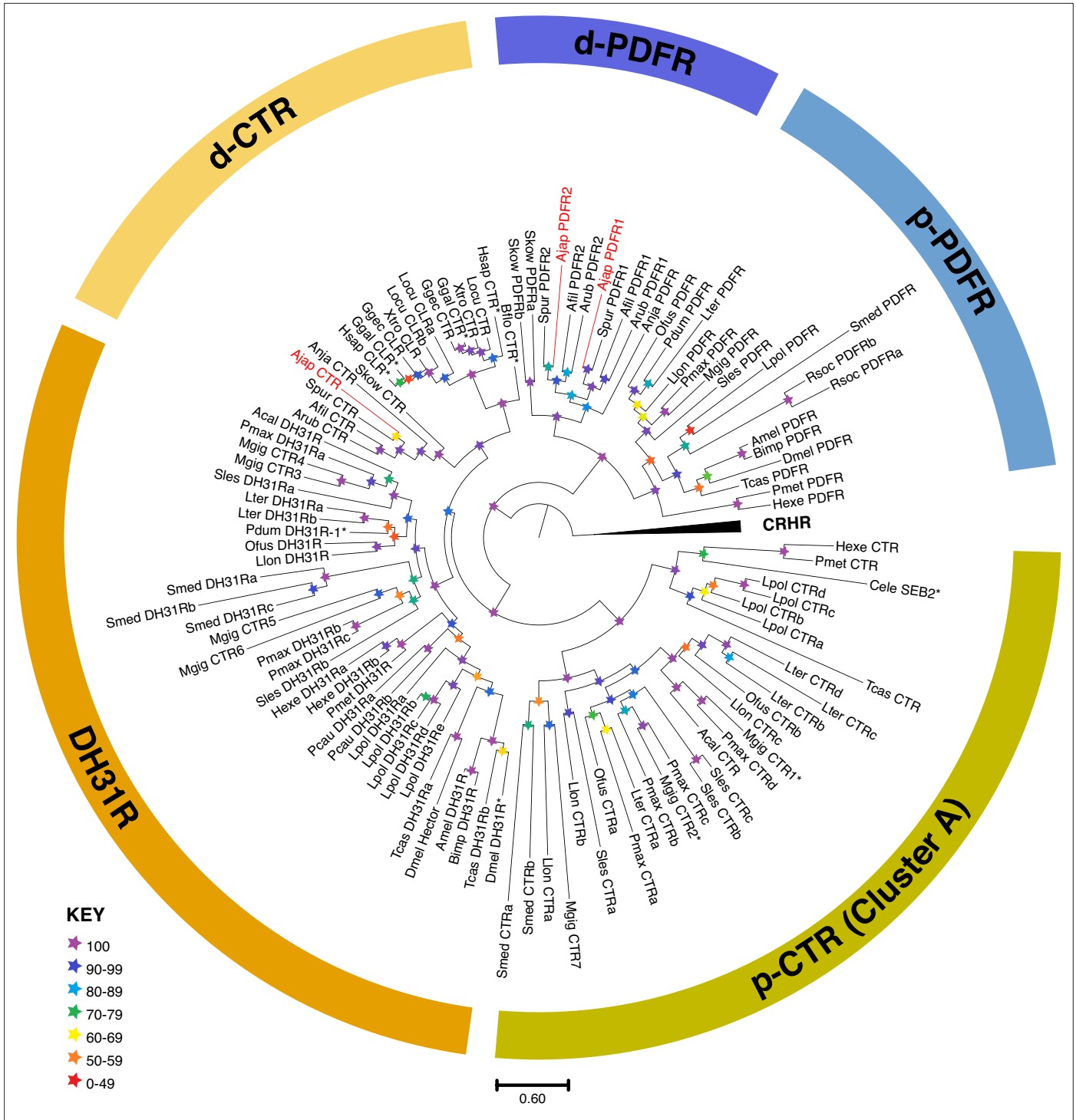

**Figure 2.** Maximum-likelihood phylogenetic tree of bilaterian CT-type and PDF-type receptor families, including those found in *A. japonicus,* rooted with CRH-type receptors. The tree comprises distinct CT-type receptor and PDF-type receptor clades with bootstrap support >90. The CT-type receptor clade is subdivided into a deuterostome-specific clade (dCTR, yellow), which includes the *A. japonicus* CTR (AjCTR), and two protostome receptor clades. One comprises DH31 receptors (DH31R, orange), and the other comprises protostome-specific CTRs (p-CTR [Cluster A], green). The name 'Cluster A' is referenced for consistency with *Cardoso et al., 2024*. The PDF-type receptors are subdivided into protostome (p-PDFR, blue) and deuterostome (d-PDFR, purple) clades, with the latter including the two *A. japonicus* PDFRs (AjPDFR1, AjPDFR2). The stars represent branch support (bootstrap 1000 replicates, see key). The scale bar indicates amino acid substitutions per site. The *A. japonicus* receptors characterized in this paper are highlighted in red, and CT-type family receptors for which ligands have been identified experimentally in other published studies are indicated with an asterisk (*Gorn et al., 1992*; *McLatchie et al., 1998*; *Johnson et al., 2005*; *Bauknecht and Jékely, 2015*; *Sekiguchi et al., 2016*; *Schwartz*

*Figure 2 continued on next page*

*Figure 2 continued*

*et al., 2019*; *Beets et al., 2023*; *Huang et al., 2024*). Species names are abbreviated as follows: Acal (*Aplysia californica*), Afil (*Amphiura filiformis*), Ajap (*Apostichopus japonicus*), Amel (*Apis mellifera*), Anja (*Anneissia japonica*), Arub (*Asterias rubens*), Bflo (*Branchiostoma floridae*), Bimp (*Bombus impatiens*), Cele (*Caenorhabditis elegans*), Cint (*Ciona intestinalis*), Dmel (*Drosophila melanogaster*), Ggal (*Gallus gallus*), Ggec (*Gekko gecko*), Hexe (*Hypsibius exemplaris*), Hsap (*Homo sapiens*), Llon (*Lineus longissimus*), Locu (*Lepisosteus oculatus*), Lpol (*Limulus polyphemus*), Lter (*Lumbricus terrestris*), Mgig (*Magallana gigas*), Ofus (*Owenia fusiformis*), Pcau (*Priapulus caudatus*), Pdum (*Platynereis dumerilii*), Pmax (*Pecten maximus*), Pmet (*Paramacrobiotus metropolita*), Rsoc (*Rotaria socialis*), Scla (*Styela clava*), Skow (*Saccoglossus kowalevskii*), Sles (*Sepioteuthis laessoniana*), Smed (*Schmidtea mediterranea*), Spur (*Strongylocentrotus purpuratus*), Tcas (*Tribolium castaneum*), Xtro (*Xenopus tropicalis*). Sequence accession numbers are listed in *Figure 2—source data 1*.

The online version of this article includes the following source data and figure supplement(s) for figure 2:

**Source data 1.** Accession numbers of CT-type, PDF-type, and CRH-type receptor families in Bilateria.

**Figure supplement 1.** Amino acid sequences and phosphorylation sites of AjCTR, AjPDFR1, and AjPDFR2 in *A. japonicus*.

**Figure supplement 2.** Characterization of AjCT1 and AjCT2 receptors (AjCTR, AjPDFR1, and AjPDFR2).

**Figure supplement 2—source data 1.** CTR/CLR-type protein sequences in Bilateria.

**Figure supplement 2—source data 2.** PDFRs protein sequences in Bilateria.

To investigate the distribution of CT-type neuropeptides in *A. japonicus* using immunohistochemistry, we employed an antiserum to the starfish (*Asterias rubens*) CT-type neuropeptide ArCT, which has been used previously for visualization of the expression of ArCT in *A. rubens* (*Cai et al., 2018*) and a CT-type neuropeptide in the feather star *Antedon mediterranea* (AmCT; *Aleotti et al., 2022*). The antigen peptide that was used to generate the ArCT antiserum (KYNSPFGASGP-NH$_2$) shares a C-terminal FGxxGP-NH$_2$ (where X is variable) motif with both AjCT1 (FGSAGP-NH$_2$) and AjCT2 (FGSGGP-NH$_2$) and therefore antibodies to ArCT may cross-react with AjCT1 and/or AjCT2. Accordingly, the ArCT antiserum revealed immunostaining in the central nervous system (radial nerve cords and circumoral nerve ring; *Figure 7A–F*), which was abolished by pre-incubation of the antiserum with the ArCT antigen peptide (*Figure 7—figure supplement 1*). More specifically, high-magnification images revealed immunostaining was localized in the ectoneural region, but not in the hyponeural region, of the radial nerve cords, with bilaterally symmetrical immunostained cells (neuronal somata) present in the ectoneural epithelium (*Figure 7C*). Likewise, immunostained cells were revealed in the ectoneural region of the circumoral nerve ring (*Figure 7D–F*). Furthermore, immunostaining was also revealed in tentacle nerves, which emanate from the circumoral nerve ring (*Figure 7D–F*), and in several regions of the digestive system, including the pharynx/stomach (*Figure 7G*), intestine (*Figure 7H*), and cloaca (*Figure 7I*). No immunostaining was observed in the longitudinal muscles (*Figure 7A and B*).

## AjCT1 and AjCT2 causes dose-dependent relaxation of longitudinal muscle and intestine preparations from *A. japonicus*

Both AjCT1 and AjCT2 caused dose-dependent relaxation of longitudinal muscles from *A. japonicus* at $10^{-10}$–$10^{-6}$ M (*Figure 8*). The maximum relaxant effects of AjCT2 ($10^{-6}$ M) on longitudinal muscles are illustrated in *Figure 8—figure supplement 1A*. At the highest concentration tested ($10^{-6}$ M), AjCT1 and AjCT2 caused 30.07 ± 7.49% and 32.50 ± 14.49% reversal of acetylcholine (ACh)-induced contraction, respectively (*Figure 8A and B*). AjCT1 had no effect on the contractile state of intestine preparations, even at concentrations as high as $10^{-5}$ M (*Figure 8C*). Unlike AjCT1, AjCT2 caused dose-dependent relaxation of intestine preparations, with 26.33 ± 8.75% reversal of ACh-induced contraction observed at $10^{-5}$ M (*Figure 8D*). Control experiments in which ACh-induced contraction was recorded without addition of CT-type peptides showed that ACh-induced contraction of longitudinal muscles and intestinal preparations persisted for ≥15 min, confirming that relaxation effects observed could be attributed to addition of AjCT1 or AjCT2 (*Figure 8—figure supplement 1B and C*).

## AjCT2 affects feeding and growth in *A. japonicus*

To investigate the effects of AjCT1 and AjCT2 in vivo, a long-term injection experiment was carried out at concentrations of $5 \times 10^{-3}$ mg/mL (low-dose group) and $5 \times 10^{-1}$ mg/mL (high-dose group), with the injection volume calculated based on the wet weight of sea cucumbers at 1 µL/g. As shown in *Figure 9A*, there were no significant changes in weight gain rate (WGR) and specific growth rate

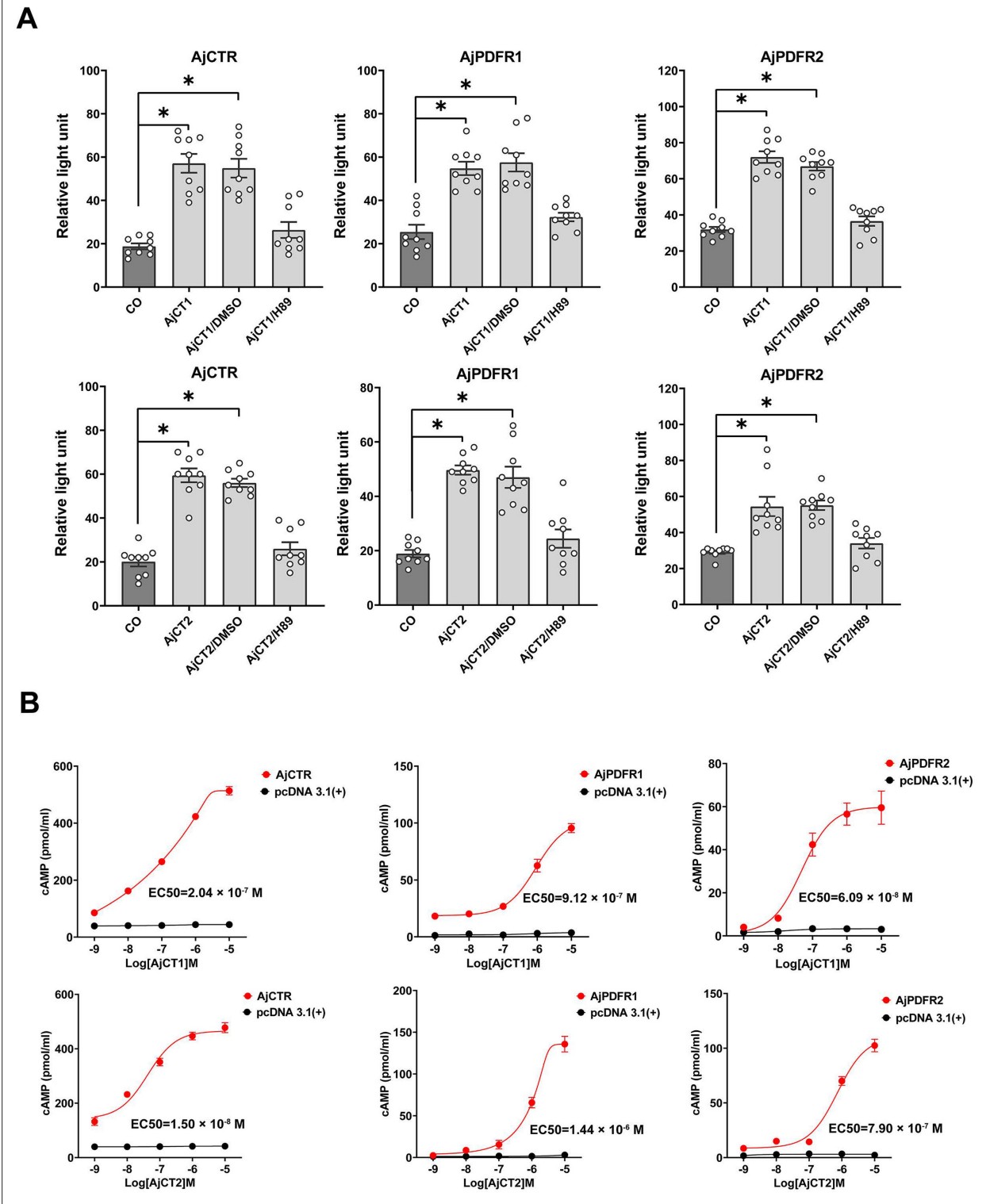

**Figure 3.** Pharmacological characterization of AjCT1 and AjCT2 as ligands for the *A. japonicus* receptors AjCTR, AjPDFR1, and AjPDFR2. (**A**) CRE-driven luciferase activity measured in HEK293T cells transfected with one of the three candidate receptors after exposure to AjCT1 or AjCT2 or neuropeptide +DMSO or neuropeptide +H89 (10 µM) or serum-free DMEM (CO, negative control). The neuropeptide concentration used here was $10^{-6}$ M. Mean values with standard deviations (n=9) are shown. * indicates a statistically significant difference with p<0.05. (**B**) Measurement of cAMP accumulation after exposure to AjCT1 or AjCT2 ($10^{-9} – 10^{-5}$ M). No cAMP elevation is observed in control experiments where HEK293T cells were transfected with empty pcDNA 3.1(+) (black circle). The estimated $EC_{50}$ of AjCT1 for AjCTR, AjPDFR1, and AjPDFR2 was $2.04 \times 10^{-7}$ M (95% CI: $1.06 \times 10^{-7}$ $–8.69 \times 10^{-7}$ M), $9.12 \times 10^{-7}$ M (95% CI: $5.56 \times 10^{-7}$ $–4.45 \times 10^{-6}$ M), and $6.09 \times 10^{-8}$ M (95% CI: $2.23 \times 10^{-8}$ $–2.84 \times 10^{-7}$ M), respectively. The estimated $EC_{50}$ of

*Figure 3 continued on next page*

*Figure 3 continued*

AjCT2 for AjCTR, AjPDFR1, and AjPDFR2 was $1.50 \times 10^{-8}$ M (95% CI: $1.03 \times 10^{-9}$ –$3.89 \times 10^{-8}$ M), $1.44 \times 10^{-6}$ M (95% CI: $1.04 \times 10^{-6}$ –$1.99 \times 10^{-6}$ M), and $7.90 \times 10^{-7}$ M (95% CI: $5.73 \times 10^{-7}$ –$1.06 \times 10^{-6}$ M), respectively. Error bars represent SEM for three independent experiments.

The online version of this article includes the following source data for figure 3:

**Source data 1.** CRE-driven luciferase activity data for *Figure 3A* and quantitative data of intracellular cAMP levels for *Figure 3B*.

(SGR) in the AjCT1 low concentration group (CT1L), AjCT1 high concentration group (CT1H), and AjCT2 low concentration group (CT2L; $p \geq 0.05$), but in the AjCT2 high concentration group (CT2H), there was a significant increase in WGR and SGR in comparison with the control group (CO; $p < 0.05$). Correspondingly, the mass of feces increased and the residual bait mass decreased (*Figure 9—figure supplement 1*). Furthermore, the relative intestinal expression levels of several growth factors were investigated, and the results showed that *AjGDF-8*, a growth inhibitory factor (*Li et al., 2016*), was significantly down-regulated in all the experimental groups. In contrast, the relative expression levels of *AjIgf* and *AjMegf6*, growth-promoting factors (*Deribe et al., 2009*; *Párrizas and LeRoith, 1997*), were significantly up-regulated in intestines (*Figure 9B*).

Subsequently, the relative intestinal expression levels of AjCT1 and AjCT2 receptors were further tested for the CT2H group. The results showed that the relative expression levels of all three receptors were significantly up-regulated, with the ranked fold change being *AjCTR >AjPDFR1>AjPDFR2* (*Figure 9C*). Conversely, the relative expression levels of the three receptor genes were significantly down-regulated in *siAjCTP1/2*-1-treated animals (*Figure 10A, B and C*). The interference rate for si*AjCTP1/2*-1 was 66% (*Figure 10—figure supplement 1*).

In long-term AjCT2 loss-of-function studies, the si*AjCTP1/2*-1 group exhibited significant reductions in WGR and SGR compared to the control group (siNC; $p < 0.05$, *Figure 11A*). Correspondingly, except for phase I, the si*AjCTP1/2*-1 group showed significantly increased remaining bait and decreased excrement across phases II-VI (*Figure 11B*). Furthermore, the growth inhibitory factor *AjGDF-8* was significantly up-regulated and the growth promoting factor *AjMegf6* was significantly down-regulated in the si*AjCTP1/2*-1 group (*Figure 11C*).

## AjPDFR2 mediates CT-type signaling-induced feeding and growth promotion in *A. japonicus*

The differential activation of signaling pathways and associated physiological outcomes between AjCT2 and AjCT1 suggested that AjCT2 may promote feeding and growth via AjPDFR2-mediated mechanisms. To investigate this hypothesis, we performed long-term knockdown experiments targeting AjPDFR2. As shown in *Figure 12A*, si*AjPDFR2-1* effectively knocked down the expression of *AjPDFR2*. In the si*AjPDFR2-1* group, both WGR and SGR were significantly decreased compared to the siNC group (*Figure 12B*). Correspondingly, the remaining bait weight was significantly increased in the si*AjPDFR2-1* group (except during phase I), and the weight of excrement was significantly decreased in phases V and VI (*Figure 12C*). Additionally, the growth-promoting factor *AjIgf* was significantly down-regulated in the si*AjPDFR2-1* group (*Figure 12D*).

## Discussion

Previous studies have revealed that the CT-type neuropeptide $DH_{31}$ can act as a ligand for a CTR/CLR-type receptor and a pigment-dispersing factor (PDF) receptor in a protostome – the insect *D. melanogaster* (*Mertens et al., 2005*; *Shafer et al., 2008*; *Schwartz et al., 2019*). Here, we show that two CT-type peptides (AjCT1 and AjCT2) act as ligands for a CTR/CLR-type receptor and two pigment-dispersing factor (PDF) type receptors in a deuterostome – the sea cucumber *A. japonicus* (phylum Echinodermata). This is an important finding because it indicates that the ability of CT-type neuropeptides to act as ligands for both CTR/CLR-type receptors and PDF-type receptors may be an evolutionarily ancient property that originated in a common ancestor of the Bilateria. Alternatively, the ability of CT-type neuropeptides to act as ligands for PDF-type receptors in *D. melanogaster* and *A. japonicus* may have evolved independently. Further studies on a wider variety of both protostome (e.g. molluscs, annelids) and deuterostome taxa (e.g. other echinoderms, hemichordates) are needed to address this issue. Here, we also investigated the signal transduction pathways that are activated

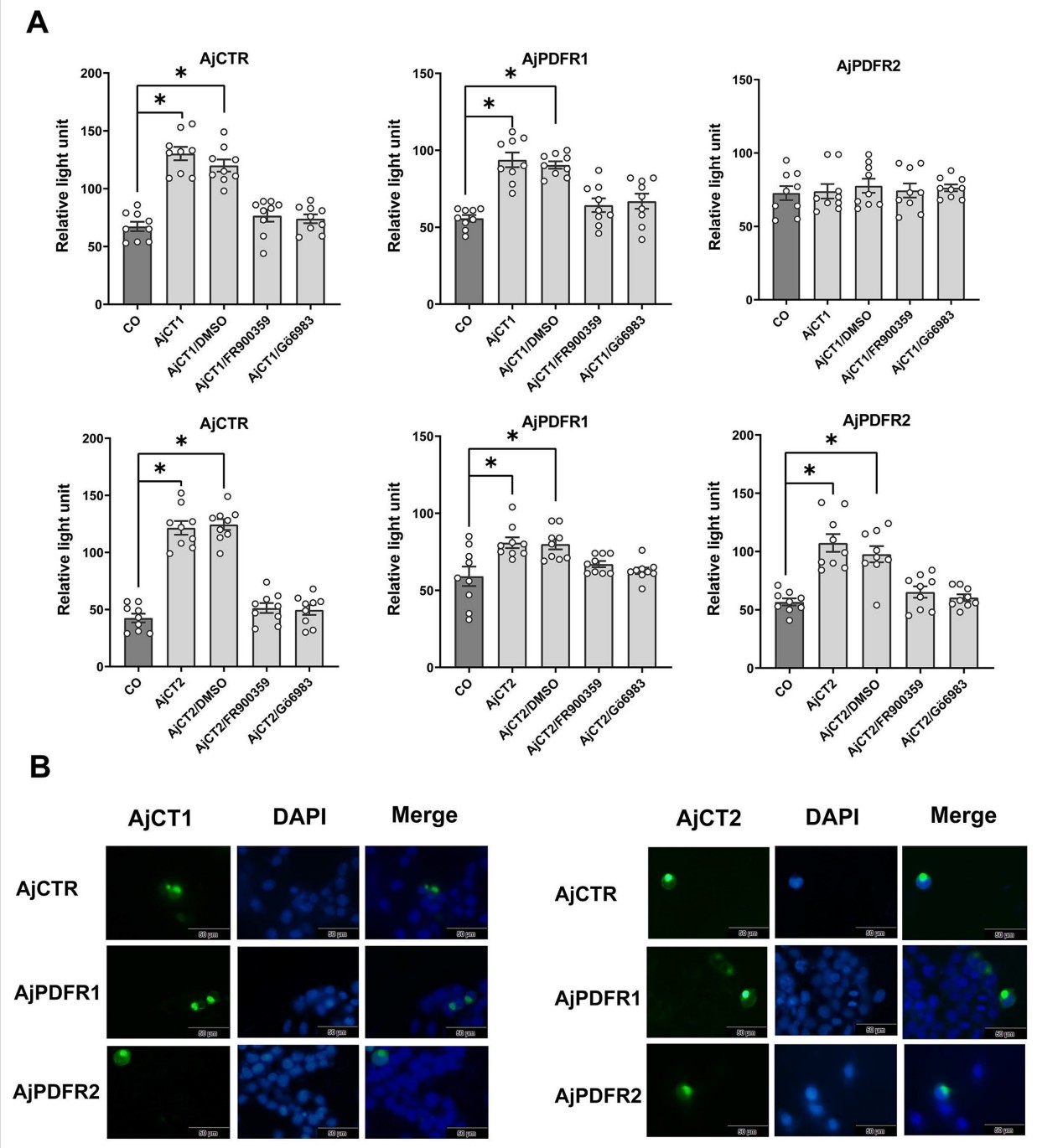

**Figure 4.** SRE-driven luciferase activity and receptor internalization in cells transfected with AjCTR, AjPDFR1, or AjPDFR2 and exposed to AjCT1 or AjCT2. (**A**) SRE-driven luciferase activity measured after incubation with neuropeptide ($10^{-6}$ M AjCT1 or AjCT2)/DMSO, neuropeptide ($10^{-6}$ M)/FR900359 (1 µM) and neuropeptide ($10^{-6}$ M)/Gö 6983 (1 µM). Mean values with standard deviations (n=9) are shown. * indicates a statistically significant difference with $p<0.05$. (**B**) Internalization of AjCTR, AjPDFR1, and AjPDFR2 after 15 min treatment with $10^{-6}$ M AjCT1 or AjCT2. Green fluorescence represents the localization of pEGFP-N1/receptors in cells. The cell nucleus probe (DAPI) was used for cell nucleus staining. Scale bar: 50 µm. All pictures are representative of three independent experiments.

The online version of this article includes the following source data and figure supplement(s) for figure 4:

**Source data 1.** SRE-driven luciferase activity data for *Figure 4A*.

**Source data 2.** Raw images of internalization for *Figure 4B*.

**Figure supplement 1.** Negative control for analysis of AjCTR, AjPDFR1, and AjPDFR2 internalization.

**Figure supplement 1—source data 1.** Raw images of negative control for internalization.

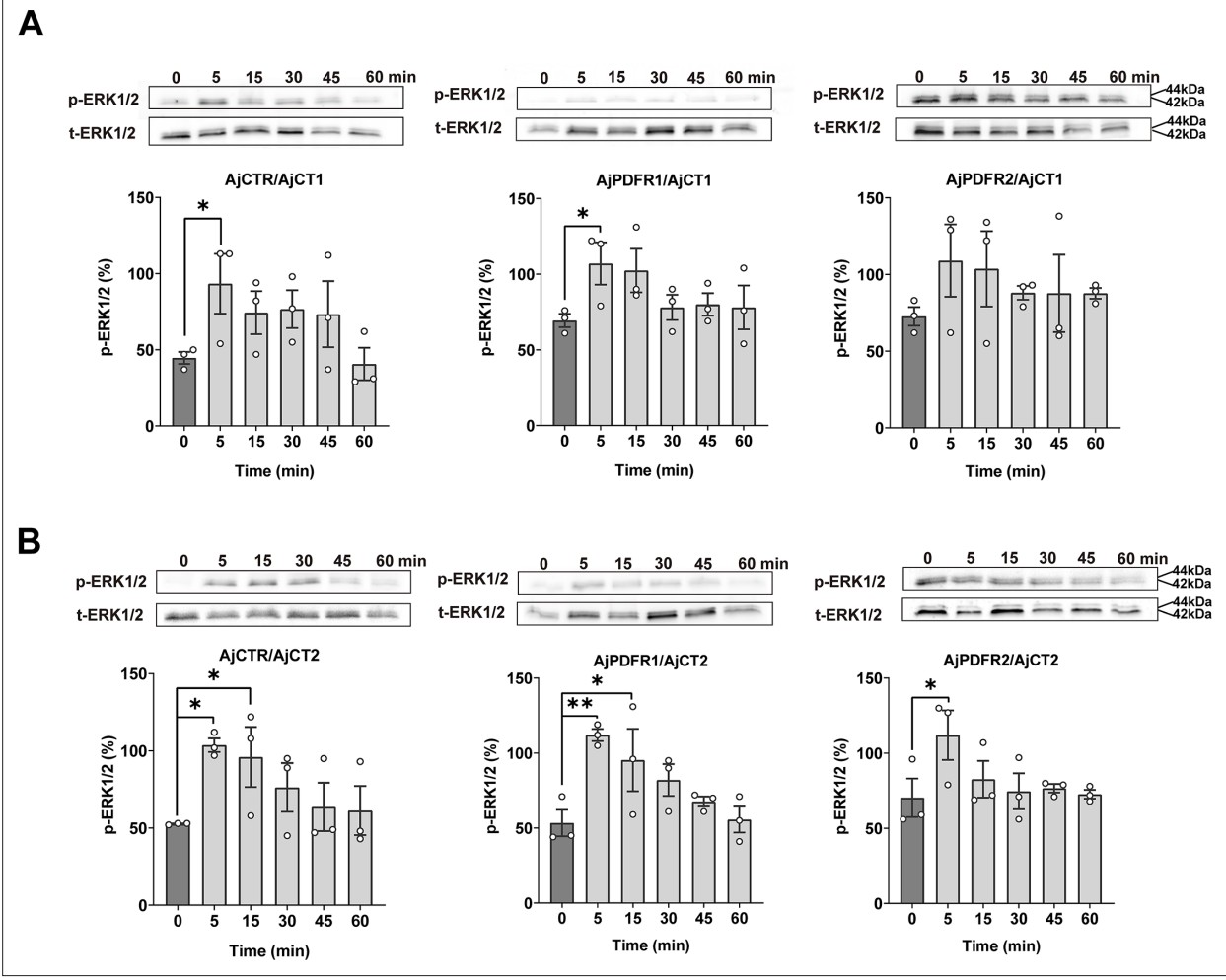

**Figure 5.** ERK1/2 activity in cells transfected with AjCTR, AjPDFR1, or AjPDFR2 and exposed to AjCT1 or AjCT2. (**A**) and (**B**) show the immunoblot intensity of representative phosphorylation bands. The transfected HEK293T cells were incubated with AjCT1 ($10^{-6}$ M) or AjCT2 ($10^{-6}$ M) for 0, 5, 15, 30, 45, and 60 min. The p-ERK1/2 was normalized based on t-ERK1/2. All images are representative of three independent experiments. Error bars represent SEM for three independent experiments. Statistically significant differences are indicated as follows: *: $p<0.05$, **: $p<0.01$.

The online version of this article includes the following source data and figure supplement(s) for figure 5:

**Source data 1.** Original data of phosphorylation levels.

**Source data 2.** PDF file containing original western blots for *Figure 5*, indicating the relevant bands.

**Source data 3.** Original files for western blot analysis displayed in *Figure 5*.

**Source data 4.** Files containing original western blots for *Figure 5*, indicating the relevant bands.

**Figure supplement 1.** Effects of PKA or PKC inhibitor on AjCT1 or AjCT2 stimulated ERK1/2 phosphorylation in AjCTR, AjPDFR1, or AjPDFR2 expressing HEK293T cells.

**Figure supplement 1—source data 1.** Original data of phosphorylation levels.

**Figure supplement 1—source data 2.** PDF file containing original western blots for *Figure 5—figure supplement 1*, indicating the relevant bands.

**Figure supplement 1—source data 3.** Original files for western blot analysis displayed in *Figure 5—figure supplement 1*.

**Figure supplement 1—source data 4.** Original files for western blot analysis displayed in *Figure 5—figure supplement 1*.

**Figure supplement 1—source data 5.** Original files for western blot analysis displayed in *Figure 5—figure supplement 1*.

**Figure supplement 1—source data 6.** Files containing original western blots for *Figure 5—figure supplement 1*, indicating the relevant bands.

**Table 1.** Activation of signaling pathways.

| Assays Receptors | cAMP/PKA/ERK1/2 | | G$\alpha$q/Ca$^{2+}$/PKC/ERK1/2 | |
|---|---|---|---|---|
| | AjCT1 | AjCT2 | AjCT1 | AjCT2 |
| *AjCTR* | ×* | ×* | √ | √ |
| *AjPDFR1* | √ | √ | √ | √ |
| *AjPDFR2* | ×* | √ | × | √ |

A tick indicates cAMP/PKA/ERK1/2 or G$\alpha$q/Ca$^{2+}$/PKC/ERK1/2 activation, the cross indicates no activation, the cross with * indicates that the cAMP/PKA pathway is activated, while the ERK1/2 pathway is not activated.

by AjCT1 and AjCT2 when these peptides bind to their receptors and then proceeded to investigate the expression and actions of these peptides to gain insights into their physiological/behavioral roles in *A. japonicus*, as discussed below.

## Alternative RNA splicing gives rise to transcripts encoding one (AjCT2) or two (AjCT1, AjCT2) CT-type neuropeptides in *A. japonicus*

In vertebrates, alternative RNA splicing gives rise to transcripts encoding CT or CGRP (*Amara et al., 1982*; *Morris et al., 1984*). Analysis of transcriptome/genome sequence data from *A. japonicus* has revealed that in this species, alternative RNA splicing gives rise to transcripts encoding one (AjCT2) or two (AjCT1, AjCT2) CT-type neuropeptides (*Rowe et al., 2014*; *Zandawala et al., 2017*; *Chen et al., 2019*; this study). A similar pattern of alternative RNA splicing has been reported for CT-type precursors in other sea cucumbers (*H. scabra* and *H. glaberrima*; class Holothuroidea) and in brittle stars (*Amphiura filiformis* and *Ophiopsila Aranea*; class Ophiuroidea) (*Zandawala et al., 2017*; *Suwan-sa-Ard et al., 2018*). In contrast, CT-type precursors identified in other echinoderm classes (Asteroidea, Echinoidea, and Crinoidea) comprise a single CT-type neuropeptide (*Rowe and Elphick, 2012*; *Mirabeau and Joly, 2013*; *Rowe et al., 2014*; *Semmens et al., 2016*; *Smith et al., 2017*; *Cai et al., 2018*; *Aleotti et al., 2022*). However, it remains to be determined if this feature of CT-type precursors in sea cucumbers and brittle stars evolved independently or originated in a common ancestor of eleutherozoan echinoderms but with subsequent loss in Asteroidea and Echinoidea (*Figure 13*).

Analysis of the sequences of AjCT1 and AjCT2 revealed the presence of two cysteine residues in their N-terminal regions, which is an evolutionarily conserved feature of CT-type peptides. Structural analysis of CT-type peptides in vertebrates and in starfish has revealed that these cysteines form a disulphide bridge in the mature peptides (*Cai et al., 2018*) and therefore it is inferred that this post-translational modification also occurs in AjCT1 and AjCT2. Furthermore, the C-terminal residue of AjCT1 and ACT2 is predicted to be an amidated proline, which is a conserved post-translational modification of CT/DH31-type peptides in many taxa, but not vertebrate CGRPs (*Cai et al., 2018*).

## Identification and characterization of receptors for CT-type neuropeptides in *A. japonicus*

Informed by previous genome-wide analysis of GPCRs in *A. japonicus* (*Huang et al., 2021*), we identified three candidate receptors for AjCT1 and/or AjCT2. Firstly, AjCTR, which is a homolog of CT/DH$_{31}$-type receptors that have been identified in other taxa, including vertebrates and insects (*Garelja et al., 2022*; *Johnson et al., 2005*; *Toribio et al., 2003*). Secondly, AjPDFR1 and AjPDF2, which are homologs of the PDF-type receptor that has been shown to be activated by DH$_{31}$ in *D. melanogaster* (*Mertens et al., 2005*; *Zandawala, 2012*).

In mammals, CTR and CLR mediate effects of all CT-related neuropeptides, but their ligand specificity relies on the presence and types of receptor activity-modifying proteins (RAMPs; *McLatchie et al., 1998*; *Purdue et al., 2002*; *Conner et al., 2004*; *Hay et al., 2018*). Thus, CTR acts as a functional receptor for CT-type peptides without an associated RAMP (*Katafuchi et al., 2009*), whereas activation of downstream signaling via CLR requires an associated RAMP (*Hay et al., 2018*). Here, we investigated the occurrence of RAMPs in *A. japonicus* using the tblastn method employed by *Sekiguchi et al., 2016*, which enabled identification of RAMPs in the cephalochordate *B. floridae*. However, no RAMPs were identified in *A. japonicus*, suggesting that AjCTR may be functionally more

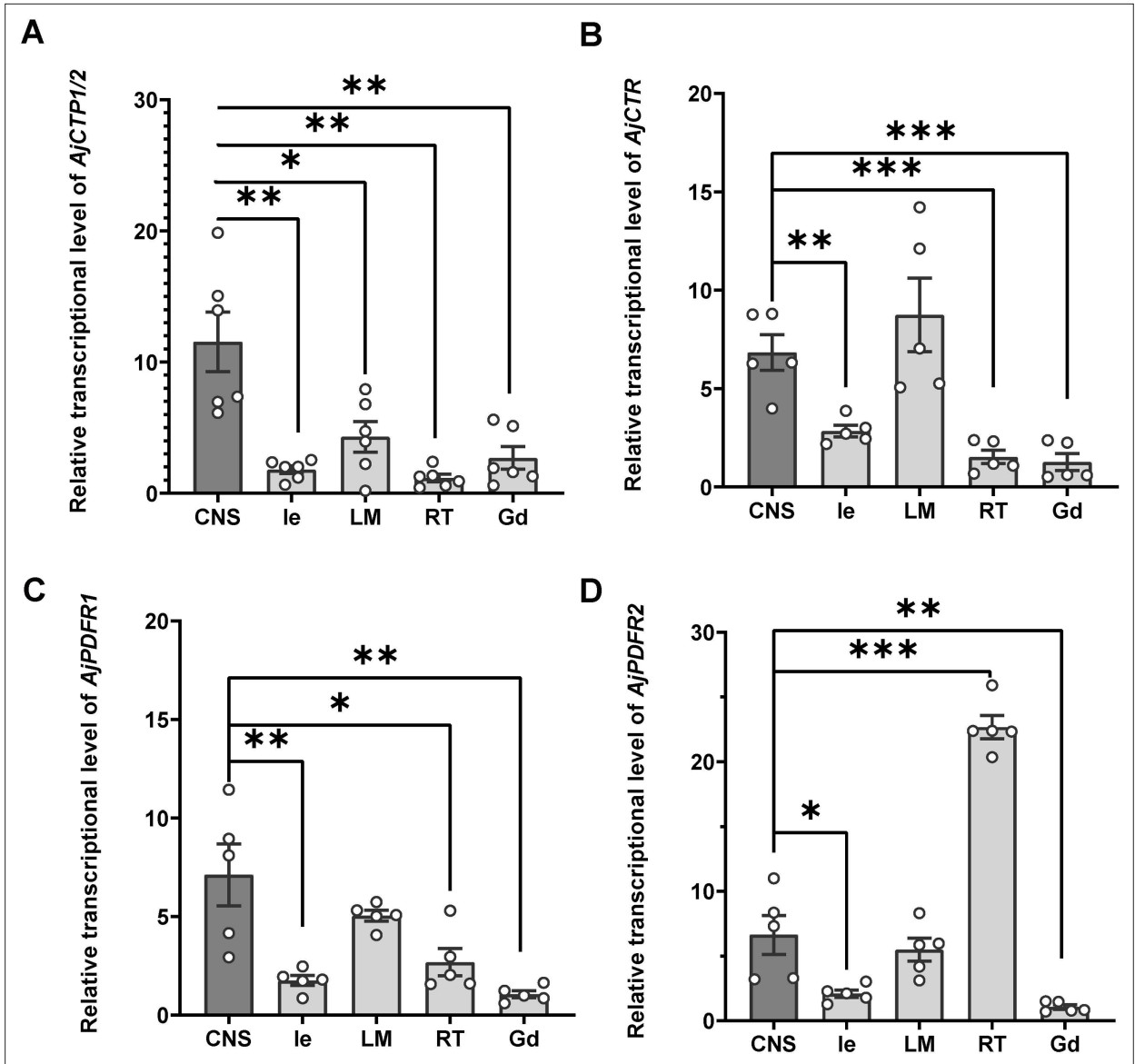

**Figure 6.** The relative expression level of *AjCTP1/2* (**A**), *AjCTR* (**B**), *AjPDFR1* (**C**), and *AjPDFR2* (**D**) in different tissues. The circumoral nervous system was selected as the control group. Values are means ± SEM (*AjCTP1/2*, n=6; *AjCTR, AjPDFR1,* and *AjPDFR2*, n=5). Asterisks represent significant differences as follows: *: $p<0.05$, **: $p<0.01$, ***: $p<0.001$. CNS, circumoral nervous system; Gd, gonad; Ie, intestine; LM, longitudinal muscle; RT, respiratory tree.

The online version of this article includes the following source data for figure 6:

**Source data 1.** Primary metadata of RT-qPCR.

---

similar to CTR than CLR. Accordingly, tblastn analysis of other available echinoderm genomes (*A. rubens*, *S. purpuratus,* and *Anneissia japonica*), as well as transcriptome databases for brittle stars (*O. victoriae*, *O. aranea,* and *A. filiformis*), also failed to identify RAMPs. Likewise, RAMPs have also not been found in protostomes (*Schwartz et al., 2019*). Therefore, RAMPs may be unique to chordates, although interestingly, RAMPs have also not been identified in the urochordate *C. intestinalis* (*Sekiguchi et al., 2009*; *Figure 13*).

As highlighted above, here we show that the CT-type peptides AjCT1 and AjCT2 both act as ligands for a CTR/CLR-type receptor (AjCTR) and two PDF-type receptors (AjPDFR1, AjPDFR2) in *A. japonicus*. Furthermore, characterization of downstream signaling mechanisms revealed that both AjCT1 and AjCT2 trigger AjCTR-mediated cAMP accumulation, $Ca^{2+}$ mobilization, and ERK signaling. These findings are consistent with previous studies on CT-type receptors in vertebrates and in the mollusc *Magallana gigas* (*Pondel, 2000*; *Walker et al., 2010*; *Andreassen et al., 2014*; *Schwartz*

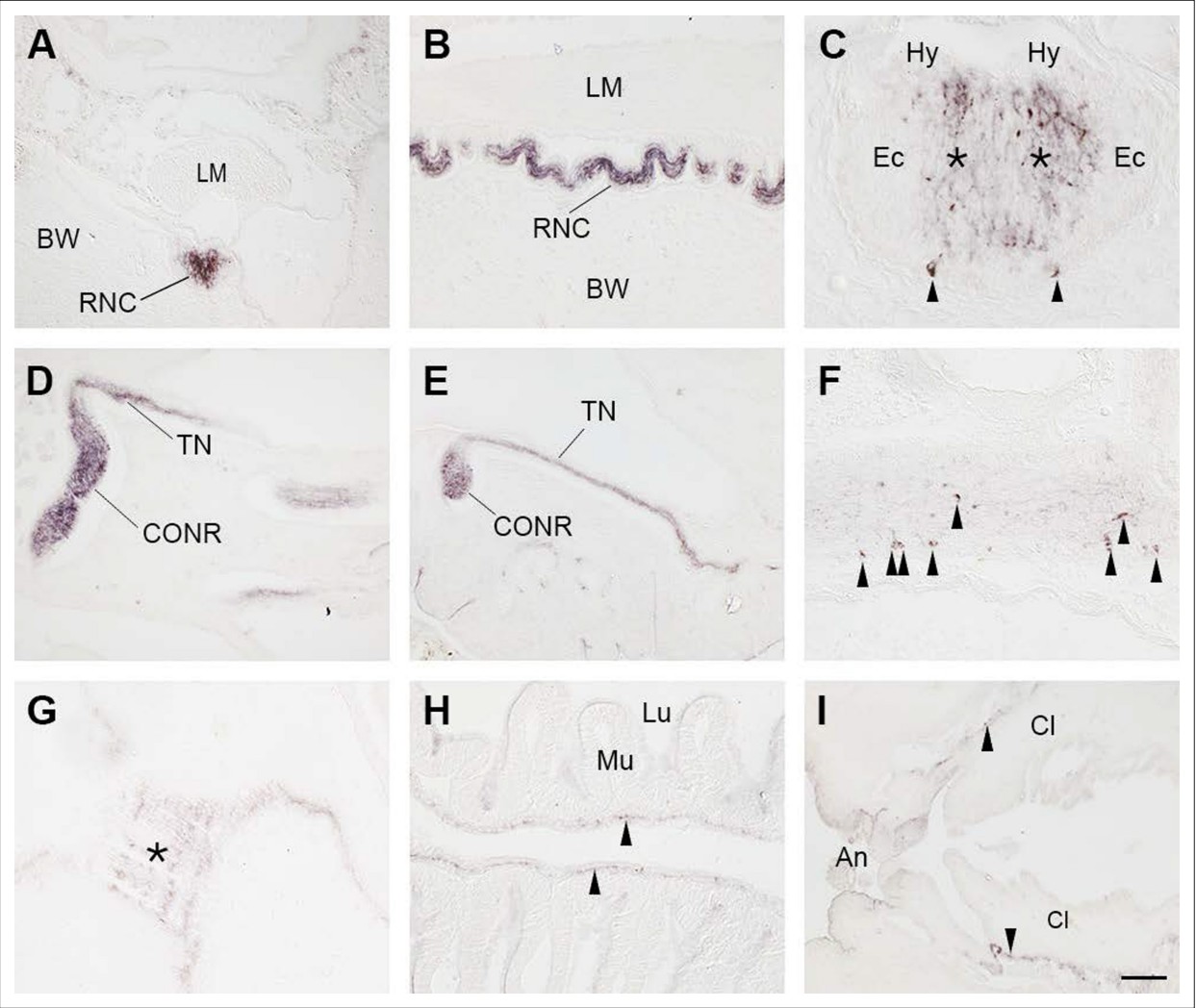

**Figure 7.** Immunohistochemical localization of CT-type neuropeptide expression in *A. japonicus* using antibodies to the starfish CT-type neuropeptide ArCT. (**A**) Transverse section showing immunostaining in the radial nerve cord. (**B**) Longitudinal section showing immunostaining in the radial nerve cord. (**C**) High-magnification image of a transverse section showing immunostained cells (arrowheads) and immunostained fibers in the neuropile (asterisks) of the ectoneural region of the radial nerve cord. Note the absence of immunostaining in the hyponeural region. (**D, E**) Longitudinal sections showing immunostaining in the circumoral nerve ring and in tentacle nerves that emanate from the circumoral nerve ring. (**F**) High-magnification image of a transverse section showing immunostained cells (arrowheads) in the circumoral nerve ring. (**G**) Longitudinal section showing immunostained fibers (asterisk) at the junction between the pharynx and the stomach. (**H**) Longitudinal section showing immunostained fibers (arrowheads) in the intestine. (**I**) Longitudinal section showing immunostained fibers (arrowheads) in the cloaca. Abbreviations: An, anus; BW, body wall; Cl, cloaca; CONR, circumoral nerve ring; Ec, Ectoneural region of the radial nerve cord; Hy, hyponeural region of the radial nerve cord; LM, longitudinal muscle; Lu, lumen; Mu, mucosa; RNC, radial nerve cord; TN, tentacle nerve. Scale bar, as shown in panel I, is 100 µm in **A**, **B**, **D**, **E**, **I**; 50 µm in **F**, **G**, **H**; 25 µm in **C**.

The online version of this article includes the following source data and figure supplement(s) for figure 7:

**Source data 1.** Raw images for immunohistochemical localization.

**Figure supplement 1.** Immunostaining in the radial nerve cord of *A. japonicus* is abolished by pre-absorption of ArCT antiserum with the ArCT antigen peptide.

**Figure supplement 1—source data 1.** Raw images of negative control for immunohistochemical localization.

*et al., 2019*). Both AjCT1 and AjCT2 trigger cAMP accumulation, $Ca^{2+}$ mobilization, and ERK signaling mediated by AjPDFR1. Both AjCT1 and AjCT2 triggered cAMP accumulation mediated by AjPDFR2, but only AjCT2 (not AjCT1) triggered $Ca^{2+}$ mobilization and ERK signaling mediated by AjPDFR2. These differences in the signal transduction mechanisms activated by AjCT1 and AjCT2 may account for differences in the pharmacological activities and physiological roles of these peptides, as discussed below.

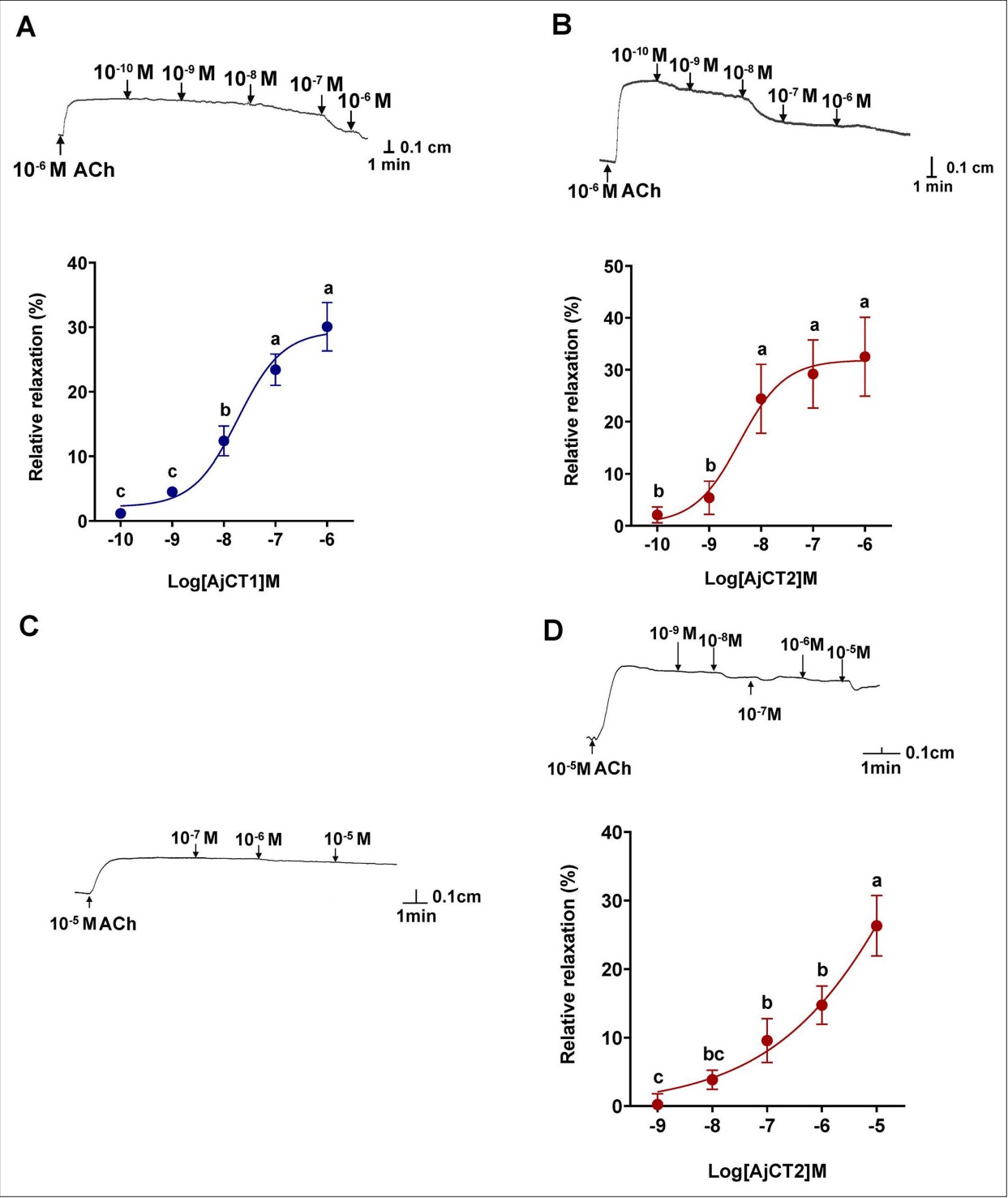

**Figure 8.** Pharmacological effects of AjCT1 and AjCT2 on longitudinal muscle and intestine preparations from *A. japonicus*. (**A**) and (**B**) show representative recordings of the relaxing effects of AjCT1 and AjCT2 on longitudinal muscle preparations. The graphs show the dose-dependent relaxing effects of AjCT1 and AjCT2 ($10^{-10}$ – $10^{-6}$ M), calculated as the percentage reversal of the contracting effect of $10^{-6}$ M ACh. (**C**) and (**D**) show representative recordings of experiments in which AjCT1 ($10^{-7}$ – $10^{-5}$ M) and AjCT2 ($10^{-9}$ – $10^{-5}$ M) were tested on intestine preparations. AjCT1 had no effect, but AjCT2 caused relaxation, and the graph shows the dose-dependent relaxing effect of AjCT2 ($10^{-9}$ – $10^{-5}$ M), calculated as the percentage reversal of the contracting effect of $10^{-5}$ M ACh. All recordings shown are representative of three independent experiments. Mean values ± SEM were determined from three preparations. Lowercase letters a, b, and c above columns indicate the statistical difference at $p < 0.05$, whilst there is no significant difference between bc and c.

*Figure 8 continued on next page*

*Figure 8 continued*

The online version of this article includes the following source data and figure supplement(s) for figure 8:

**Source data 1.** Raw images of representative recordings.

**Source data 2.** Primary metadata of relaxing efficiency for *Figure 8A, B and D*.

**Figure supplement 1.** Positive control on experimental tissues and pharmacological effects of 10⁻⁶ M AjCT2 on longitudinal muscle from *A. japonicus*.

**Figure supplement 1—source data 1.** Raw images of positive control recordings.

### AjCT1 and AjCT2 act as muscle relaxants in *A. japonicus*

Analysis of the expression of the gene encoding the precursors comprising AjCT2 or AjCT1 and AjCT2 revealed the occurrence of transcripts in the circumoral nervous system as well as in peripheral tissues/ organs, including the longitudinal muscle, digestive system, and gonads. Immunohistochemical analysis using an antiserum to the starfish (*A. rubens*) CT-type neuropeptide ArCT enabled visualization of CT-type neuropeptide(s) in juvenile specimens of *A. japonicus*. Intense and extensive immunostaining was revealed in the ectoneural region of the radial nerve cords and circumoral nerve ring, consistent with the high levels of *AjCTP1/2* transcripts detected in circumoral nervous tissue. Immunostained nerve fibers were also observed in the digestive system (including the intestine), consistent with the detection of *AjCTP1/2* transcripts in intestine. However, no immunostaining was observed in longitudinal muscles, which is inconsistent with the detection of *AjCTP1/2* transcripts in this tissue. This

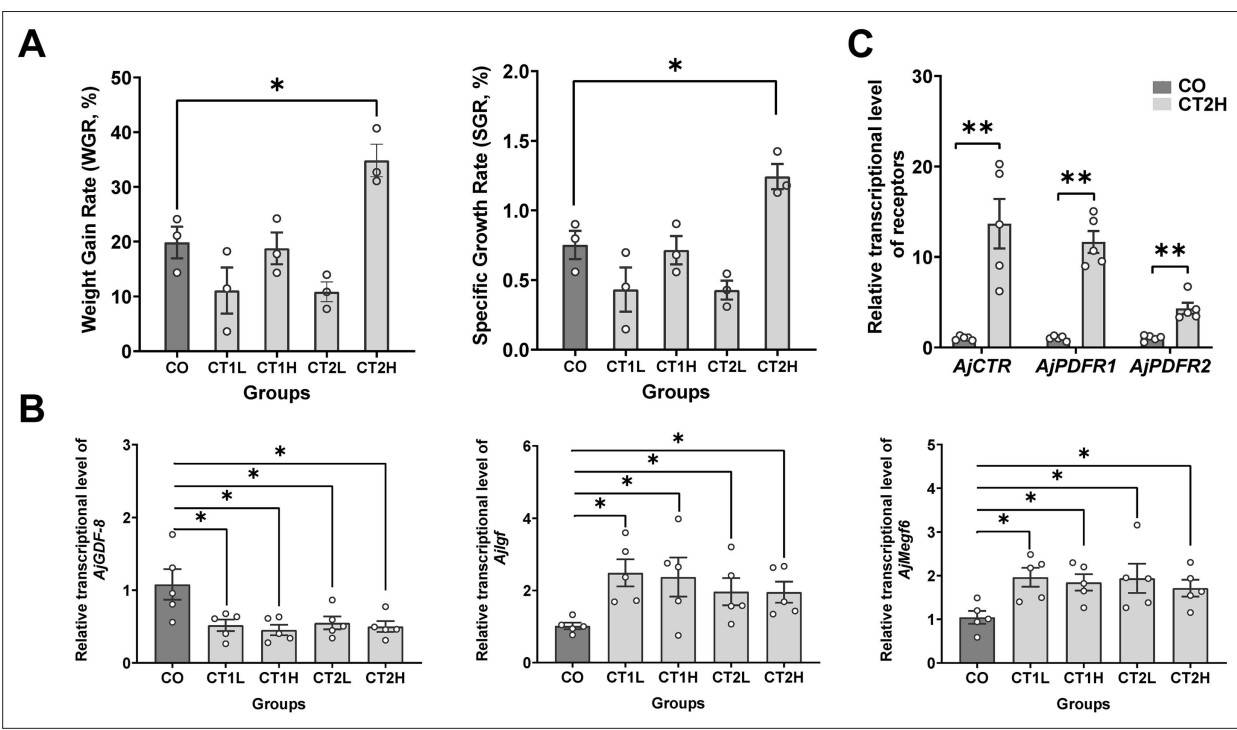

**Figure 9.** In vivo pharmacological tests revealed that AjCT1 and AjCT2 affect feeding and growth-related gene expression in *A. japonicus*. (**A**) Weight gain rate (WGR) and specific growth rate (SGR) in experimental and control groups. Values are means ± SEM (n=3, representing three parallel experiments of each group). (**B**) The relative intestinal transcript expression levels of *AjGDF-8, AjIgf,* and *AjMegf6* in different groups after injection of AjCT1 or AjCT2 or sterilized seawater (CO) *in A. japonicus*. Values are means ± SEM (n=5 biological duplication). (**C**) The relative intestinal expression levels of AjCT1/2 receptors (*AjCTR, AjPDFR1,* and *AjPDFR2*) in CT2H and CO groups after 24 days injection experiment. Values are means ± SEM (n=5 biological duplication). CO: control group (sterilized seawater); CT1L/CT2L: AjCT1/AjCT2 low concentration group (5×10⁻³ mg/mL); CT1H/CT2H: AjCT1/AjCT2 high concentration group (5×10⁻¹ mg/mL). * indicates statistically significant differences with p<0.05, ** indicates statistically significant differences with p<0.01.

The online version of this article includes the following source data and figure supplement(s) for figure 9:

**Source data 1.** Primary metadata of WGR, SGR, and RT-qPCR.

**Figure supplement 1.** Measurement of the mass of remaining bait and excrement for AjCT2-treated and control animals.

**Figure supplement 1—source data 1.** Primary metadata of remaining bait weight and excrement weight.

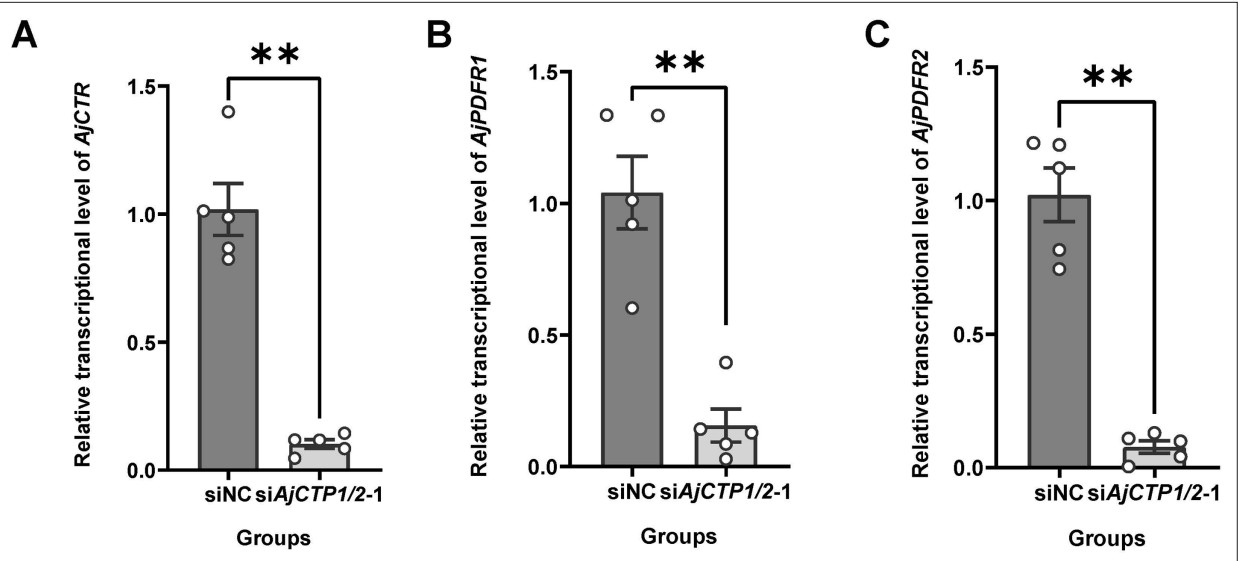

**Figure 10.** The relative intestinal expression levels of AjCT1/2 receptors after *AjCTP1/2*-1 knockdown. (**A**), (**B**), and (**C**) show the relative expression levels of *AjCTR*, *AjPDFR1*, and *AjPDFR2,* respectively. Values are means ± SEM (n=5). ** indicates statistically significant differences with p<0.01.

The online version of this article includes the following source data and figure supplement(s) for figure 10:

**Source data 1.** Primary metadata of RT-qPCR.

**Figure supplement 1.** The relative expression level of *AjCTP1/2* in siNC, si*AjCTP1/2*-1, and si*AjCTP1/2*-2.

**Figure supplement 1—source data 1.** Primary metadata of RT-qPCR results.

may reflect differences in the detection sensitivity of the techniques used. In accordance with the expression pattern of AjCTP1/2 in *A. japonicus*, analysis of CT-type neuropeptide expression in other echinoderms has revealed a widespread pattern of expression in larval and adult starfish (*A. rubens*) and in juvenile and adult feather stars (*Antedon mediterranea*; *Mayorova et al., 2016*; *Cai et al., 2018*; *Aleotti et al., 2022*).

Several CT-related peptides act as muscle relaxants in vertebrates (*Steiner et al., 1991*; *Pinto et al., 1996*; *Yoshimoto et al., 1998*; *Edvinsson et al., 2001*; *Golpon et al., 2001*; *Kandilci et al., 2008*) and, accordingly, previous studies have revealed that CT-type neuropeptides cause relaxation of muscle preparations in starfish, including locomotory organs (tube feet) and the apical muscle, which is functionally similar to the longitudinal muscle in sea cucumbers (*Cai et al., 2018*). Therefore, it was of interest to investigate if AjCT1 and/or AjCT2 act as muscle relaxants in *A. japonicus*. Both AjCT1 and AjCT2 caused dose-dependent relaxation of longitudinal muscle preparations, consistent with the relaxing effect of CT-type peptides on starfish apical muscle preparations. It is worth noting that while AjCT2 at $10^{-6}$ M exhibited the maximum efficacy in reversing ACh-induced contraction, its relaxing effect weakened compared to the $10^{-8}$ M treatment when applied at progressively higher concentrations. This diminished efficacy likely stems from receptor desensitization, induced by repeated AjCT2 application to longitudinal muscle preparations (*Tsurumaki et al., 2003*; *Arrowsmith and Wray, 2014*). Furthermore, AjCT2, but not AjCT1, caused dose-dependent relaxation of intestine preparations from *A. japonicus*. Interestingly, the CT-type peptide ArCT does not cause relaxation of cardiac stomach preparations from the starfish *A. rubens* (*Cai et al., 2018*). Therefore, from a functional perspective, ArCT is more like AjCT1 than AjCT2, and we speculate that this may reflect loss of an exon encoding an AjCT2-like peptide in the Asteroidea lineage. Further insights into this issue and the evolution of CT-type neuropeptide function in echinoderms could be obtained by investigation of the actions of CT-type peptides in brittle stars and sea urchins. Differences in the effects of AjCT1 and AjCT2 on the intestine may be explained by differences in their signal transduction mechanisms. Thus, only AjCT2, not AjCT1, triggered AjPDFR2-mediated activation of $Ca^{2+}$ mobilization and ERK signaling, and therefore we speculate that the relaxing effect of AjCT2 on the intestine may be mediated by activation of AjPDFR2.

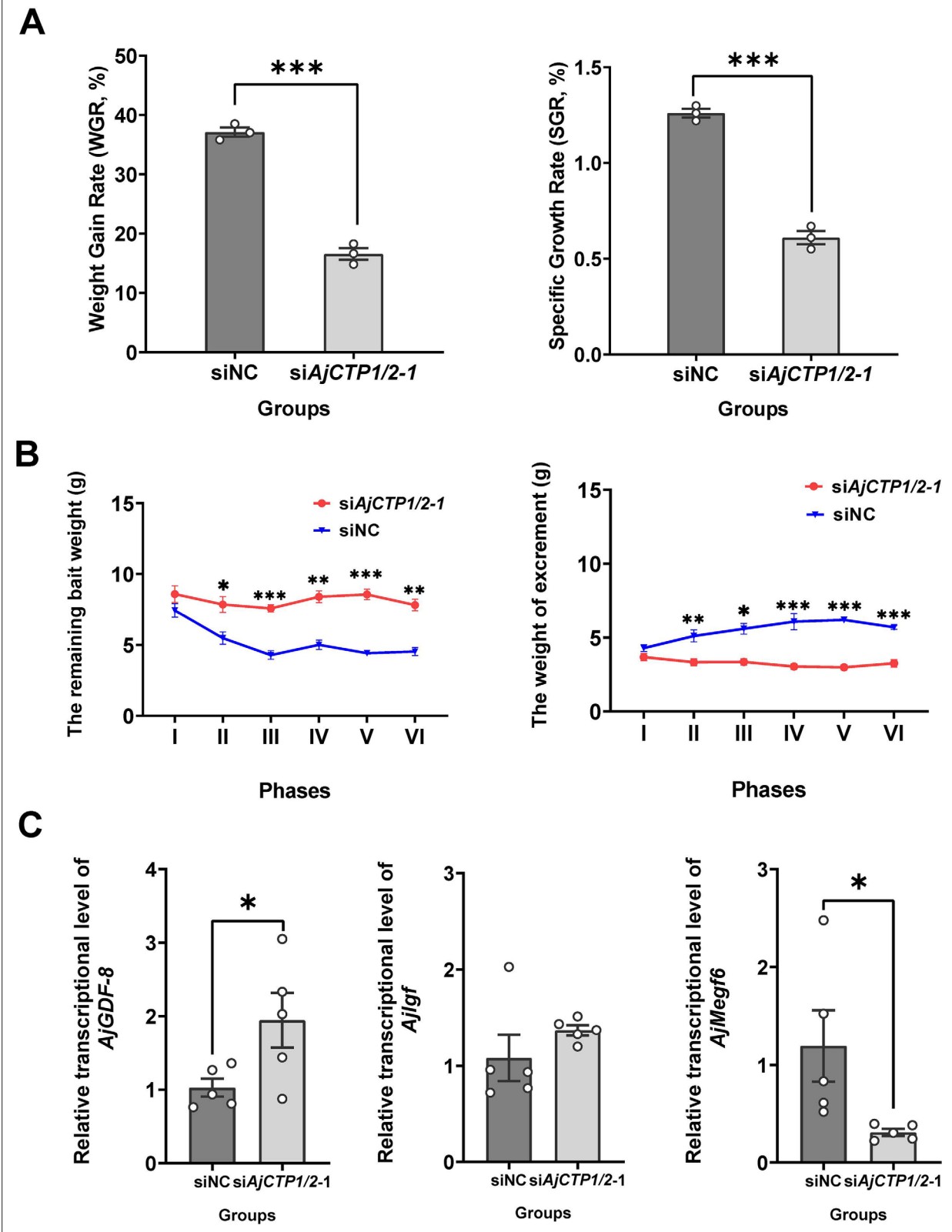

**Figure 11.** Effects of 24-day *AjCTP1/2*-1 knockdown in vivo on feeding and growth in *A. japonicus*. (**A**) Weight gain rate (WGR) and specific growth rate (SGR) in siNC and *siAjCTP1/2*-1 groups. Values are means ± SEM (n=3, representing three parallel experiments for each group). (**B**) Calculation of the remaining bait and excrement mass in different phases. The 24-day experiment was divided into six phases (**I–VI**), with each phase comprising 4 days. Values are means ± SEM (n=3, representing three parallel experiments for each group). (**C**) The relative intestinal transcript expression levels of *AjGDF-8*,

*Figure 11 continued on next page*

*Figure 11 continued*

*AjIgf*, and *AjMegf6* in siNC and *siAjCTP1/2*-1 groups. Values are means ± SEM (n=5 biological duplication). siNC: negative control group; si*AjCTP1/2*-1: effectively experimental groups targeting *AjCTP1/2*. Asterisks represent statistically significant differences as follows: *: $p < 0.05$, **: $p < 0.01$, ***: $p < 0.001$.

The online version of this article includes the following source data for figure 11:

**Source data 1.** Primary metadata of WGR, SGR, remaining bait weight, excrement weight, and RT-qPCR results.

## AjCT2/AjPDFR2 stimulates feeding and growth in *A. japonicus*

In vivo pharmacological assays demonstrated that only high concentrations of AjCT2 significantly enhanced feeding and growth rates in *A. japonicus*. In contrast, neither a low concentration of AjCT2 nor any concentration of AjCT1 (low or high) induced detectable effects. Furthermore, long-term knockdown of *AjCTP1/2* further validated the essential role of AjCT2 in regulating feeding and growth in this species (*Figure 14*). To elucidate the receptor mediating AjCT2's feeding- and growth-promoting effects, we selected AjPDFR2 based on its distinct activation profile: AjCT2 selectively activated AjPDFR2, inducing downstream ERK1/2 phosphorylation, whereas AjCT1 exhibited no activity toward this receptor. Given this receptor specificity, we performed *AjPDFR2* knockdown experiments, which revealed phenotypic changes consistent with those in *AjCTP1/2* knockdown animals, including significantly reduced WGR and SGR, alongside increased remaining bait accumulation and diminished excrement output compared to control. Collectively, these results support a model wherein AjCT2 promotes feeding and growth in *A. japonicus* via AjPDFR2-dependent activation of the cAMP/PKA/ERK1/2 and $G\alpha q/Ca^{2+}$/PKC/ERK1/2 cascades. Considering the inherent complexity of neuropeptide signaling systems, which involve multiple GPCR subtypes coupled to diverse signaling cascades, ligands bound to the same receptor may activate distinct G protein subforms within a single cell (*Hartmann, 2003*; *Mendel et al., 2020*). Receptor activation modes may be modulated by structural polymorphisms or binding site diversity (*Hung Choy Wong et al., 2000*; *Changeux, 2010*), as well as by the differential efficacy of peptides in activating receptors in vivo.

Collectively, these findings indicate that AjCT2 has a physiological role in stimulating feeding and growth in *A. japonicus* (*Figure 14*). To the best of our knowledge, this is the first study to report orexigenic and growth-promoting effects of a CT-type neuropeptide in bilaterian animals. Interestingly, previous studies have reported the expression of CT-type neuropeptides in the digestive system of invertebrate chordates (*Sekiguchi et al., 2009*; *Sekiguchi et al., 2016*). However, in contrast to the stimulatory effect of AjCT2 on feeding/growth in *A. japonicus*, CT-type neuropeptides have been found to inhibit feeding and growth in vertebrates (*Perlow et al., 1980*; *Del Prete et al., 2002*; *Nakamura et al., 2018*; *Mathiesen et al., 2020*). Thus, it appears that opposing roles of CT-type neuropeptide signaling systems as regulators for feeding/growth have evolved in deuterostomes, with a stimulatory role reported here in an echinoderm and an inhibitory role reported previously in vertebrates (*Figure 13*). Further research is now needed to investigate if the stimulatory effect of AjCT2 on feeding/growth observed here in *A. japonicus* also applies to other sea cucumber species. Moreover, our discovery of this effect of AjCT2 on *A. japonicus* may provide a basis for development of novel approaches to improve aquaculture of this economically important species.

## Materials and methods

### Key resources table

| Reagent type (species) or resource | Designation | Source or reference | Identifiers | Additional information |
|---|---|---|---|---|
| sequence-based reagent | 5'-CGCGGATCCAAAAT GCAGGAGAACGATTC-3' 5'-CCGGAATTCTTAAA CTACCGTTGTTTTAT-3' | Custom synthesized by Tsingke | | Oligonucleotide primers for pcDNA3.1(+) plasmid construction-- AjCTR |
| Cell line (*Homo sapiens*) | Human Embryonic Kidney cells 293T (HEK293T) | Procell | CL-0005 RRID:CVCL_0063 | |
| Antibody | Anti-phospho-ERK (T202/Y204) (rabbit monoclonal) | Cell Signaling Technology | Cat#: 4370 RRID:AB_2315112 | Used for Western Blotting (1:2000) |

*Continued on next page*

*Continued*

| Reagent type (species) or resource | Designation | Source or reference | Identifiers | Additional information |
|---|---|---|---|---|
| Antibody | anti-ERK (rabbit monoclonal) | Cell Signaling Technology | Cat#: 4695 RRID:AB_390779 | Used for Western Blotting (1:1000) |
| Antibody | HRP Conjugated Goat anti-Rabbit IgG h+l antibody (Goat polyclonal) | Absin | abs20040 RRID:AB_2938713 | Used for Western Blotting (1:2000) |
| Antibody | (Rabbit polyclonal) antibodies to *Asterias rubens* calcitonin-type neuropeptide ArCT | https://doi.org/10.3389/fnins.2018.00382 | RRID:AB_2721239 | Used for immunohistochemistry (1:LRE1000 - 1:PDF4000) |
| Antibody | Peroxidase-AffiniPure Goat Anti-Rabbit IgG (H+L) conjugated to Horseradish Peroxidase (Goat polyclonal) | Jackson ImmunoResearch | RRID:AB_2313567 Cat. no. 111–035-003 | Used for immunohistochemistry (1:1000) |
| Peptide, recombinant protein | SCSNKFAGCAHMKVANAVLKQNSRGQQQFKFGSAamide | Sangon biotech | | AjCT1 neuropeptide with a disulfide bond and C-terminal amidation |
| Peptide, recombinant protein | RVGGCGDFSGCASLKAGRDLVRAMLRPSKFGSGGPamide | Sangon biotech | | AjCT2 neuropeptide with a disulfide bond and C-terminal amidation |
| Chemical compound, drug | FR900359 | Cayman Chemical | Item No. 33666 CAS No. 107530-18-7 | $G_{\alpha}q$ protein inhibitor |
| Chemical compound, drug | Gö 6983 | Cayman Chemical | Item No. 13311 CAS No. 133053-19-7 | PKC inhibitor |
| Chemical compound, drug | H89 | Absin | abs810011 CAS No. 130964-39-5 | PKA inhibitor |
| commercial assay or kit | SMARTer RACE 5'/3' Kit | Takara | Cat No. 634858 | |
| commercial assay or kit | FastPure Gel DNA Extraction Mini Kit | Vazyme | DC301-01 | |
| commercial assay or kit | Hifair III 1st Strand cDNA Synthesis SuperMix | YEASEN | Cat No. 11141ES60 | |
| commercial assay or kit | Hieff UNICON Universal Blue qPCR SYBR Green Master Mix | YEASEN | Cat No. 11184ES08 | |
| commercial assay or kit | Firefly luciferase reporter gene assay kit | MK | MF4001 | |
| commercial assay or kit | cAMP Assay kit | R&D systems | Cat No. KGE002B | |
| Recombinant DNA reagent | pcDNA 3.1+vector with neomycin selectable marker (mammalian expression vector) | YouBio | VT1001 | |
| Recombinant DNA reagent | pEGFP-N1 vector with neomycin selectable marker (mammalian expression vector) | YouBio | VT1110 | |
| Software, algorithm | LabChart | ADInstruments | Version 8.0.7 RRID:SCR_017551 | |
| Software, algorithm | Prism | GraphPad | Version 8.0 RRID:SCR_002798 | |
| Software, algorithm | ImageJ | http://rsb.info.nih.gov/ij | Version 1.0 RRID:SCR_003070 | |
| Software, algorithm | Illustrator | Adobe | RRID:SCR_010279 | |

*Continued on next page*

*Continued*

| Reagent type (species) or resource | Designation | Source or reference | Identifiers | Additional information |
|---|---|---|---|---|
| Software, algorithm | IQ-tree2 | https://iqtree.github.io/ | Version 2.0.7 for linux 64-bi RRID:SCR_017254 | |
| Software, algorithm | MUSCLE5 | https://www.drive5.com/muscle5/ | RRID:SCR_011812 version 5.1 for linux64 | Algoritm PPP: https://drive5.com/muscle5/Muscle5_SuppMat.pdf |
| Software, algorithm | trimAl | http://trimal.cgenomics.orghttps://github.com/inab/trimal/tree/trimAl | Version: 1.2rev59 for linux64 RRID:SCR_017334 | http://trimal.cgenomics.org/publications. https://doi.org/10.1093/bioinformatics/btp348 |

## Animals

Adult sea cucumbers were collected from an aquaculture breeding farm in Weihai, Shandong, China. All the animals were acclimated in breeding aquaria (18°C, 32 ppt) for at least 7 days and fed daily with a commercially formulated diet. Tissues used for cDNA cloning and qRT-PCR were obtained from six adult *A. japonicus* (~120 g) and kept at –80°C until use. Juvenile specimens of *A. japonicus* (~15 mm in length) were used for immunohistochemical analysis of CT-type neuropeptide expression. In vitro functional analysis and in vivo experiments were carried out using adult sea cucumbers with body masses of ~120 g and ~20 g, respectively.

## Ethical approval

All methods were carried out in accordance with the ARRIVE guidelines (https://arriveguidelines.org) for animal experiments. All animal care and use procedures were approved by the Institutional Animal Care and Use Committee of Ocean University of China (Permit Number: 20141201). They were performed according to the Chinese Guidelines for the Care and Use of Laboratory Animals (GB/T 35892-2018).

## Molecular structure and phylogenetic analysis of CT-type neuropeptides and precursors

*A. japonicus* CT-type neuropeptide precursor sequences (*AjCTP1*, GenBank accession number: MF401985; *AjCTP2*, GenBank accession number: MF401986) were identified in our previous study (*Chen et al., 2019*). Based on the published sequences, transcript-specific primers for *AjCTP1* and *AjCTP2* (*Supplementary file 1*) were designed using Primer3Plus (https://www.primer3plus.com/) and sequence accuracy was confirmed by SMARTer RACE 5′/3′ Kit (TaKaRa, Kusatsu, Japan) according to the manufacturer's instruction. A sequence translation tool (https://web.expasy.org/translate/) was used to obtain the corresponding amino acid sequences encoded by the cloned cDNAs. To identify the conserved features of CT-type neuropeptides in Bilateria, sequences were aligned using Clustal-X with default settings and then predicted using the online tool Multiple Sequence Comparison Display Tool (http://www.bio-soft.net/sms/index.html). To investigate relationships of AjCTP1, AjCTP2, and CT-type precursors from other Bilateria, phylogenetic analyses were performed using the Neighbor-Joining method. The precursor sequences were aligned using the MUSCLE plugin (iterative, 10 iterations, UPGMB as clustering method), a phylogenetic tree was generated using the neighbor joining method with MEGA7 (v.7170509) and bootstrap values were calculated from 1000 replicate analysis and then modified using iTOL (https://itol.embl.de/login.cgi). The accession numbers or citations for all sequences used for analysis are listed in *Figure 1—source data 1*.

## Cloning, sequence analysis, and phylogenetic tree construction for candidate AjCT1/2 receptors in *A. japonicus*

Five potential receptors for AjCT1 and AjCT2 in *A. japonicus* (GenBank accession number: PIK49998.1, PIK49999.1, PIK49604.1, PIK34007.1, and PIK62458.1) were identified based on genome-wide analysis (*Huang et al., 2021*). Sequence comparison revealed that both PIK49998.1 and PIK62458.1 were partial sequences of PIK34007.1. Thus, three candidate receptors belonging to Class B (Secretin-like)

GPCRs named as *AjCTR* (Calcitonin receptor, GenBank accession number: PIK49604.1), *AjPDFR1* (Pigment-dispersing factor receptor 1, GenBank accession number: PIK34007.1), and *AjPDFR2* (Pigment-dispersing factor receptor 2, GenBank accession number: PIK49999.1) were predicted and then analyzed. To confirm the accuracy of these sequences, cDNAs were cloned using the SMARTer RACE 5'/3' Kit (Takara, Cat No. 634858) and then sequenced (Beijing Genomics Institute, China).

Phylogenetic relationships of AjCTR, AjPDFR1, and AjPDFR2 with GPCRs belonging to the CT-type (including DH31 and Cluster A) and PDF-type receptor families in other bilaterian animals were investigated, with CRH-type receptors used as an outgroup. Receptor sequences (see *Figure 2—source data 1*) were aligned using MUSCLE 5.1 with the PPP algorithm (*Edgar, 2022*). The alignment was trimmed with the gappyout option using trimAl 1.2rev59 (*Capella-Gutiérrez et al., 2009*), and maximum-likelihood phylogenetic trees were generated with IQ-tree 2.0.7 for Linux 64-bit using the SH-aLRT and ultrafast bootstrap (UFBoot) methods with nearest neighbor interchange (NNI) correction tests (*Guindon et al., 2010*; *Guindon and Gascuel, 2003*; *Hoang et al., 2018*; *Minh et al., 2020*; *Nguyen et al., 2015*). The VT+F+I+G4 amino acid substitution model was selected, with branch support of 1000 bootstrap replicates. The tree was rooted using CRH-type receptors as the outgroup and edited using Treeviewer version 2.2.0 (*Bianchini and Sánchez-Baracaldo, 2024*).

In addition, the online tool NetPhos-3.1 (https://services.healthtech.dtu.dk/service.php?NetPhos-3.1) was used to predict potential phosphorylation sites. The conserved domains of protein sequences were predicted by the Batch SMART plugin of TBtools (Toolbox for biologists, v.1.098652) software. For analysis of gene synteny, neighboring genes of *AjCTR, AjPDFR1,* and *AjPDFR2* were found based on an *A. japonicus* chromosome-level genome assembly (*Wang et al., 2022*) and the corresponding genes in the starfish *Asterias rubens* and the sea urchin *Lytechinus variegatus* were also identified by performing BLASTp in the NCBI database. Specific primers for cloning of cDNAs encoding the candidate receptors are shown in the *Supplementary file 1* and the details of the sequences used for constructing the phylogenetic tree are listed in *Figure 2—source data 1*.

## Testing AjCT1 and AjCT2 as ligands for candidate receptors and exploration of downstream signaling pathways

AjCT1 and AjCT2 (both with a disulfide bridge between the two cysteine residues and with C-terminal amidation) were custom synthesized by GL Biochem Ltd. (Shanghai, China) with >95% purity. To test candidate receptors for CT-type neuropeptides and investigate their downstream activation mechanisms, the following assays were used to measure cAMP signaling or $Ca^{2+}$ signaling: (i). direct measurement of cAMP levels using ELISA (ii). indirect measurement of cAMP signaling using a cAMP response element-Luciferase (CRE-Luc) assay that measures cAMP-induced gene expression, (iii). serum response element (SRE)-Luc assays that measure $Ca^{2+}$-induced gene expression, (iv). receptor translocation assays and (v). Western blotting.

This study utilized the HEK293T cell line (Catalog No. CL-0005, RRID:CVCL_0063), obtained from Procell (Wuhan Pricella Biotechnology Co., Ltd.). The cell line was authenticated by short tandem repeat (STR) profiling, which matched the reference profile for HEK293T, and was confirmed to be free of mycoplasma contamination through a service provided by ATCC (https://www.procell.com.cn/p/293t-hek-293t-cl-0005-68219). *AjCTR, AjPDFR1,* and *AjPDFR2* incorporating enhanced green fluorescent protein (EGFP) were constructed respectively according to the method described by *Li et al., 2022*. Briefly, the complete ORFs of candidate receptors were cloned and sequenced. Then, the PCR products were digested with EcoRI and BamHI (TaKaRa, Kusatsu, Japan) and then incorporated into the pEGFP-N1 vector (YouBio, Changsha, China) that was subject to the same double-digestion. Finally, the constructed plasmid with correct sequences was transfected into HEK293T cells seeded on glass coverslips in 12-well plates. After incubation with AjCT1 ($10^{-6}$ M), AjCT2 ($10^{-6}$ M) or serum-free DMEM (Biosharp, Hefei, China) for 15 min, cells were washed with PBS (Biosharp, Hefei, China) and fixed with 4% PFA for 10 min. Finally, cell nuclei were stained with DAPI (G-CLONE, Beijing, China) for 5 min, and an Olympus bright field light microscope was used for image capture. The pcDNA 3.1(+) vectors (YouBio, Changsha, China) comprising the complete ORF of the three candidate receptors were also constructed as described by *Li et al., 2022*. Then, they were co-transfected with the cAMP reporter pCRE-Luc (YouBio, Changsha, China) or $Ca^{2+}$ signaling reporter pSRE-Luc plasmid including a PKC reporter gene (YouBio, Changsha, China) into HEK293T cells. Incubation was carried out with AjCT1 ($10^{-6}$ M), AjCT2 ($10^{-6}$ M) or serum-free DMEM (CO, negative control) for 4 hr at

37°C. Finally, luciferase activity was tested using the firefly luciferase assay kit (MK, Shanghai, China). When required, transfected cells described above were co-incubated with H89 (PKA inhibitor; Absin, Shanghai, China), FR900359 (Gαq protein inhibitor; Cayman Chemical, Ann Arbor, Michigan), or Gö 6983 (PKC inhibitor; Cayman Chemical, Ann Arbor, Michigan) before the experiment to explore the potential signaling pathways. To confirm the concentration of intracellular cAMP after stimulation, HEK293T cells expressing pcDNA 3.1(+)/*AjCTR*, pcDNA 3.1(+)/*AjPDFR2,* or pcDNA 3.1(+)/*AjPDFR1* were treated with AjCT1 or AjCT2 ($10^{-9} - 10^{-5}$ M), respectively, for 15 min after 2 h starvation in serum-free medium. Subsequently, cAMP was measured using an ELISA-based cAMP assay kit (R&D Systems, USA and Canada) following the manufacturer's instruction. HEK293T cells transfected with empty pcDNA3.1(+) were used as a negative control. The specific primers used for constructing plasmids are listed in *Supplementary file 1*.

Western blotting was performed as described previously (*Li et al., 2022*). Briefly, HEK293T cells transfected with the constructed pcDNA 3.1(+) plasmid were starved in serum-free medium for 2 hr and then incubated with AjCT1 or AjCT2 ($10^{-6}$ M). PVDF membranes containing proteins were blocked with 5% BSA (G-CLONE, Beijing, China) and incubated overnight with antibodies to phosphorylated ERK1/2 kinases (CST, Boston, China). After that, goat anti-rabbit horseradish peroxidase-conjugated secondary antibody (Absin, Shanghai, China) was applied and incubated, followed by the addition of Omni-ECL Femto Light Chemiluminescence reagents (EpiZyme, Shanghai, China). Finally, RVL-100-G (ECHO, America) was applied for visualization of the PVDF membrane. When required, transfected cells were co-incubated with PKA or PKC inhibitor before the experiment to identify signaling pathways that activate downstream ERK1/2 cascade.

## Total RNA extraction and quantitative real-time PCR (qRT-PCR) detection

qRT-PCR was employed to detect *AjCTR*, *AjPDFR1*, *AjPDFR2*, *AjCTP1*, and *AjCTP2* transcript expression in five tissues from *A. japonicus*, including the intestine, longitudinal muscle, respiratory tree, gonad, and circumoral nervous system, and the circumoral nervous system was selected as the control group. Because it was difficult to design specific primers for *AjCTP1* and *AjCTP2* (*Zheng et al., 2022*), total transcript expression levels for *AjCTP1/2* (*AjCTP1 + AjCTP2*) were investigated. Total RNA was extracted from the collected tissues using Trizol RNA isolation reagent (Vazyme, Nanjing, China) and Hifair III 1st Strand cDNA Synthesis SuperMix (YEASEN, Shanghai, China) was used to remove genomic DNA and synthesize cDNA according to manufacturer's instructions. For qRT-PCR, amplification was performed in 20 µL volume reactions based on Hieff UNICON Universal Blue qPCR SYBR Green Master Mix (YEASEN, Shanghai, China) and expression levels were assessed by a Corbett Rotoe-Gnen Q (QIAGEN, Germany). *β*-actin (GenBank accession number: PIK61412.1) and *β*-tubulin (GenBank accession number: PIK51093) were used as housekeeping genes for normalization (*Zhao et al., 2014*). The $2^{-\Delta\Delta CT}$ method was applied to analyze the relative expression levels. Specific sequences of primers used for qRT-PCR are listed in *Supplementary file 1*.

## Localization of CT-type neuropeptide expression in *A. japonicus* using immunohistochemistry

Juvenile specimens of *A. japonicus* (~15 mm in length) were fixed for 24 hr at room temperature or for 48 hr at 4°C in Bouin's fixative (75 mL saturated picric acid in seawater, 25 mL 37% formaldehyde, 5 mL acetic acid) days and then washed in water before being transferred to 50% ethanol followed by 70% ethanol. The fixed specimens were wrapped in tissue paper soaked in 70% ethanol and then transported from China to the UK. Following dehydration through 90% and 100% ethanol and clearing in xylene, specimens were embedded in paraffin wax and then sectioned (9 or 10 µm) transversely or longitudinally using a Leica RM2145 microtome. Sections were mounted on chrome alum-gelatin coated glass slides and then after incubation in xylene to remove wax followed by 100% ethanol, slides were incubated in 1% hydrogen peroxide in phosphate buffered saline (PBS, pH 7.3) to quench endogenous peroxidases. After using an ethanol series to rehydrate sections, slides were blocked in 5% goat serum in PBS containing 0.1% Tween-20 (PBST). Then slides were incubated overnight at 4°C with the ArCT antiserum diluted to 1:2000, 1:3000, or 1:4000 in PBST. Following washing in PBST, slides were incubated with Peroxidase-AffiniPure Goat Anti-Rabbit IgG (H + L) conjugated to Horseradish Peroxidase (RRID:AB_2313567; Jackson ImmunoResearch, West Grove, PA) diluted 1:1000 in

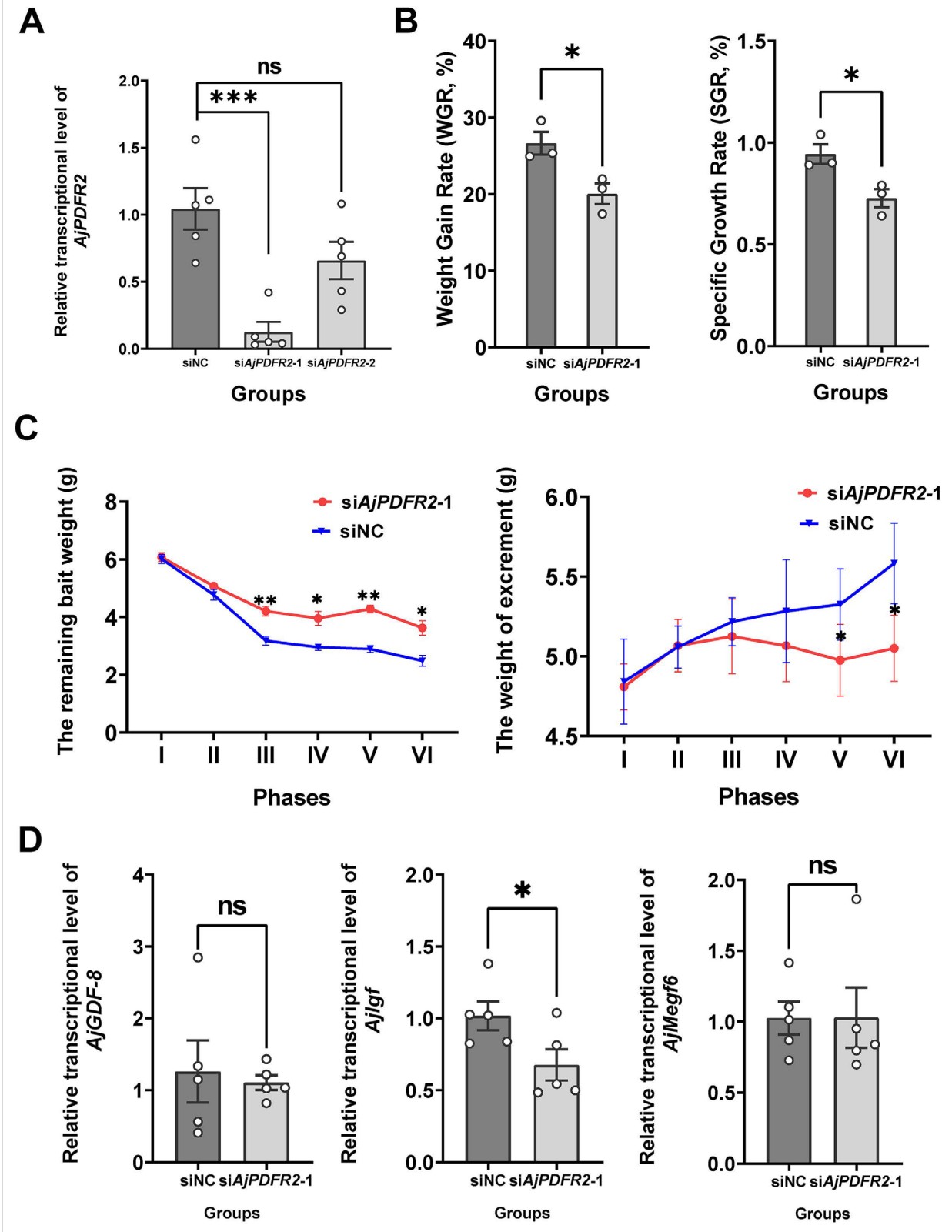

**Figure 12.** Effects of *AjPDFR2*-1 knockdown in vivo on feeding and growth in *A. japonicus*. (**A**) The relative expression level of *AjPDFR2* in siNC, si*AjPDFR2*-1, and si*AjPDFR2*-2. (**B**) Weight gain rate (WGR) and specific growth rate (SGR) in siNC and *siAjPDFR2*-1 groups. Values are means ± SEM (n=3, representing three parallel experiments for each group). (**C**) The remaining bait and excrement weight in different phases. The 24-day experiment was divided into six phases (I–VI), with each phase comprising 4 days. Values are means ± SEM (n=3, representing three parallel experiments of each

*Figure 12 continued on next page*

*Figure 12 continued*

group). (**D**) The relative intestinal transcript expression levels of *AjGDF-8, AjIgf,* and *AjMegf6* in siNC and *siAjPDFR2*-1 groups. Values are means ± SEM (n=5 biological duplication). siNC: negative control group; siA*jPDFR2*-1 and siA*jPDFR2*-2: experimental groups targeting A*jPDFR2*. Asterisks represent statistically significant differences as follows: *: $p < 0.05$, **: $p < 0.01$, ***: $p < 0.001$. ns: not significant.

The online version of this article includes the following source data for figure 12:

**Source data 1.** Primary metadata of WGR, SGR, remaining bait weight, excrement weight, and RT-qPCR results.

2% goat serum/PBST for 3 hr. Bound antibodies were revealed using a solution of 0.05% diaminobenzidine (VWR Chemicals, UK), 0.05% nickel chloride (Sigma-Aldrich, Gillingham, UK), 0.015% hydrogen peroxide (VWR Chemicals, UK) diluted in PBS. When staining was observed, slides were washed in water and dehydrated through an ethanol series into xylene and mounted with coverslips over DPX mounting medium (Thermo Fisher Scientific). The specificity of immunostaining was assessed by pre-absorption of the ArCT antiserum (at 1:100 dilution) for 1 hr at room temperature with the ArCT antigen peptide (25 μM) prior to testing on sections at 1:3000 dilution. Images of stained sections were taken using a QIClich CCD Colour Camera (Qimaging, UK) linked to a DMRAI light microscope (Leica Microsystems, UK) with Volocity v. 6.3.1 image analysis software (Perkin-Elmer, USA) running on an iMac computer (27 inch with OS Yosemite, v. 10.10). Montages of images were prepared and labeled using GIMP (version 3.0.2) and Adobe Photoshop (version 26.7) running on a MacBook Pro laptop (OS Sequoia 15.5).

## Analysis of the in vitro activity of AjCT1 and AjCT2 on preparations from *A. japonicus*

Longitudinal muscles and intestines (20 mm in length) from *A. japonicus* were dissected and then suspended in a 50 mL glass organ bath containing 20 mL sterilized seawater. Both ends of the preparations were tied with cotton ligatures and then one end was tied to a metal hook and the other end was connected with a High Grade Isotonic Transducer (ADinstruments MLT0015, Oxford, UK). Before testing, the preparation was stabilized for about 20 min, and the resting tension was adjusted to 1.0 g. Informed by previous findings from *Cai et al., 2018*, we investigated if AjCT1 and AjCT2 caused relaxation of preparations. First, acetylcholine (ACh) was applied to induce the contraction of preparations, and then when a stable state of contraction was achieved, 20 μL AjCT1 or AjCT2 at a range of concentrations was added and relaxation responses were recorded. Output signal (cm) from PowerLab data acquisition hardware (ADinstruments PowerLab 4/26, Oxford, UK) was calculated using LabChart (v8.0.7) software.

## Investigation of the in vivo pharmacological effects of AjCT1 and AjCT2 in *A. japonicus*

A total of 150 adult sea cucumbers were randomly assigned to five groups (30 individuals per group), including two high concentration groups (CT1H and CT2H, $5 \times 10^{-1}$ mg/ml), two low concentration groups (CT1L and CT2L, $5 \times 10^{-3}$ mg/ml), and the control group (CO, sterilized seawater), with each group consisting of three parallel experiments comprising 10 animals. Before the experiments, sea cucumbers were starved for a day to normalize their physiological status, and their wet weights were measured. AjCT1 and AjCT2 were dissolved in sterilized seawater and then injected through the peristome at noon every other day (1 μl/g wet weight), and the CO group was injected in the same way but with sterilized seawater (1 μl/g wet weight). Feces and residual bait were collected and weighed after drying every day. After 24 days, wet weights of sea cucumbers were measured again. Then, WGR and SGR were calculated to assess their feeding and growth conditions. Formulas are as follows: $WGR = 100*(W_t-W_0)/W_0$, $SGR = 100*(lnW_t-lnW_0)/D$. In the formula, $W_0$ represents the initial wet body weight (g), $W_t$ represents the final wet body weight (g), D represents the feeding days. Subsequently, animals were dissected and intestines were collected for detecting the expression levels of the three receptor genes (*AjCTR, AjPDFR1, AjPDFR2*) and several growth factors, including multiple epidermal growth factor 6 (*AjMegf6*, GenBank accession number: MG018199.1), insulin-like growth factor (*AjIgf*, GenBank accession number: PIK50518.1) and growth and differentiation factor-8 (*AjGDF-8*, GenBank accession number: KP100064.1; *Li et al., 2016*; *Wang et al., 2018*). The specific primers for qRT-PCR analysis are listed in the *Supplementary file 1*.

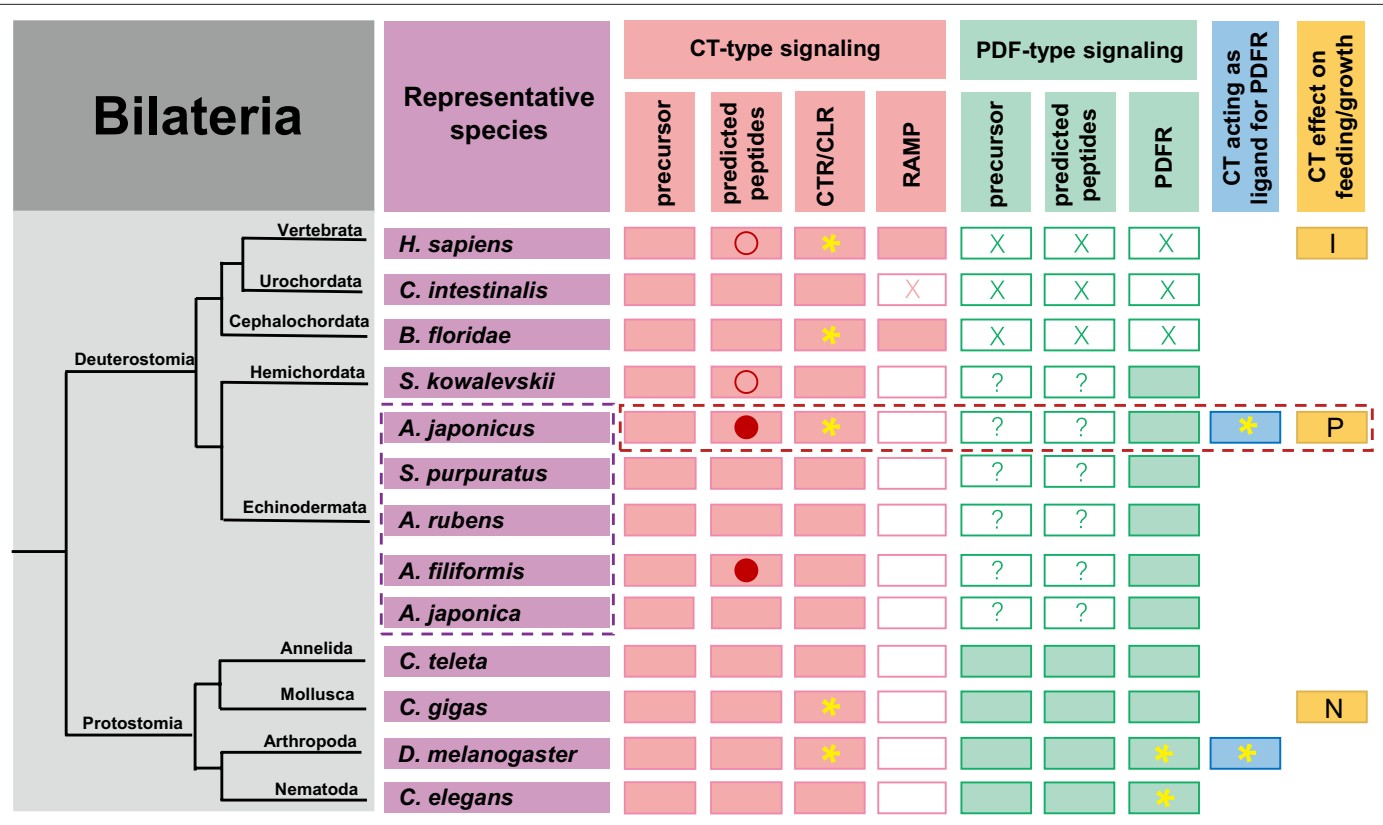

**Figure 13.** Phylogenetic summary of the occurrence and properties of CT-type and PDF-type signaling systems and effects of CT-type peptides on feeding/growth regulation in representative species. Echinoderms are boxed in purple with dashed lines, and *A. japonicus*, the focus of this study, is framed in red with dashed lines. Pink filled rectangles indicate the occurrence of CT-type precursors/neuropeptides, CTR/CLR-type receptors, and RAMPs. Green filled rectangles indicate the occurrence of PDF-type precursors/neuropeptides and receptors. Blue filled rectangles indicate species in which CT-type peptides have been shown to act as ligands for PDF-type receptors. Yellow filled rectangles indicate species in which effects or lack of effects of CT-type peptides on feeding/growth have been reported (I: Inhibition; P: Promotion; N: No effect). Red circles indicate that neuropeptides are generated by alternative splicing of precursor transcripts, with an unfilled circle indicating that transcripts encode one neuropeptide and with a filled circle indicating that transcripts can encode two neuropeptides in tandem. A yellow asterisk marks species in which neuropeptides have been shown to act as ligands for receptors experimentally. Unfilled rectangles indicate the absence of a protein. Unfilled rectangles containing a cross (**X**) indicate loss of a protein during evolution. Unfilled rectangles containing a question mark (?) indicate the existence of precursors/peptides that have yet to be identified.

## Loss-of-function experiments by RNA interference (RNAi)

To investigate how receptor expression is affected by knock-down of *AjCTP* expression, an RNAi experiment was carried out. It was difficult to design *AjCTP1*- and *AjCTP2*-specific interference oligonucleotides, and therefore, we designed and customized generic interference oligonucleotides targeting both *AjCTP1* and *AjCTP2* (si*AjCTP1/2*-1 and si*AjCTP1/2*-2, containing the extra 2' ome modification; Beijing Tsingke Biotech Co., Ltd. China). Meanwhile, a negative control siRNA (siNC) was also customized for the experiment (*Supplementary file 1*), which does not target genes in *A. japonicus*. The si*AjCTP1/2*-1, si*AjCTP1/2*-2, or siNC dry powder was diluted to 20 μM with DEPC water following the manufacturer's instruction, then 60 μl sterilized seawater was mixed with 20 μl si*AjCTP1/2*-1, si*AjCTP1/2*-2, or siNC, and 20 μl Lipo6000 transfection reagent (Beyotime, Shanghai, China) as the transfection solution. Eighteen sea cucumbers were randomly divided into three groups and injected with three different transfection solutions respectively. The mixed solutions were injected through the peristome once a day (1 μl/g wet weight), and intestines were dissected and frozen in liquid nitrogen 3 days after injection. The expression level of *AjCTP1/2* was investigated by qRT-PCR, and the interference rate was calculated using $(1 - 2^{-\Delta\Delta CT})$ to determine the inhibition efficiency of siRNAs. Finally, the relative expression levels of *AjCTR*, *AjPDFR1*, and *AjPDFR2* were investigated using the selected *A. japonicus* intestines.

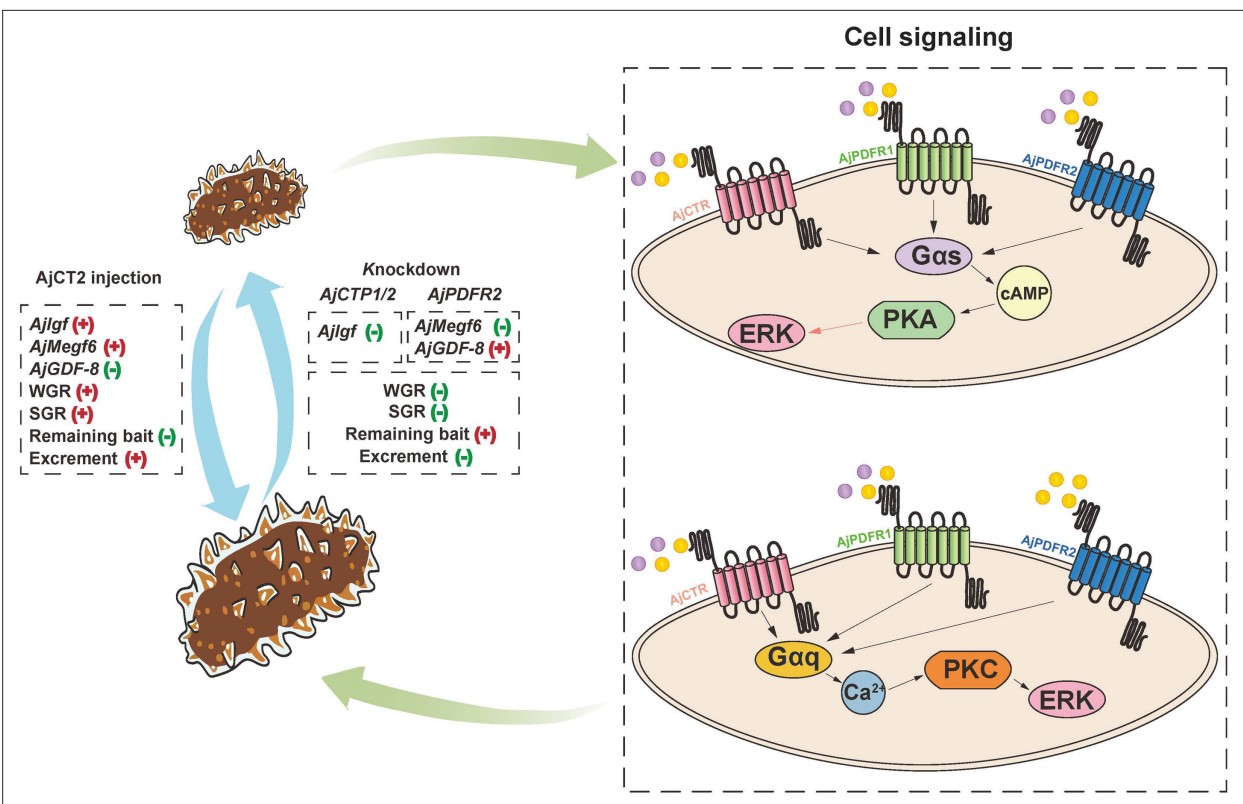

**Figure 14.** A schematic showing proposed molecular mechanisms by which CT-type signaling regulates feeding and growth in *A. japonicus*. AjCT1 and AjCT2 are represented by purple and yellow circles, respectively. For cell signaling, black arrows indicate that this process is activated, while the pink arrow indicates that ERK1/2 is activated in the AjCT1/AjPDFR1, AjCT2/AjPDFR1, and AjCT2/AjPDFR2 signaling pathways.

To assess the effects of CT-type signaling inhibition on feeding and growth in *A. japonicus*, a 24-day RNAi experiment was conducted following the methodology outlined in Section (Investigation of the in vivo pharmacological effects of AjCT1 and AjCT2 in *A. japonicus*). Sea cucumbers were randomly allocated into two groups: the negative control group received siNC injections, while the experimental group was injected with si*AjCTP1/2*-1 (effectively targeting *AjCTP1/2*). Throughout the experiment, the dry weight of remaining bait and excrement, along with the wet weight of individual sea cucumbers, was recorded. Subsequently, WGR and SGR, as indices of feeding efficiency and growth performance, were calculated. In addition, the expression levels of the three growth factors (*AjMegf6*, *AjIgf*, and *AjGDF-8*) in the intestinal tissue were tested by qPCR.

To investigate the potential mediation of CT-type signaling effects on feeding and growth by AjPDFR2 in *A. japonicus*, an RNAi experiment targeting *AjPDFR2* was performed following the same knockdown protocol established for *AjCTP*. Briefly, the effective siRNA targeting *AjPDFR2* (si*AjPDFR2*-1 or si*AjPDFR2*-2) was selected by testing *AjPDFR2* expression in the intestines of injected sea cucumbers using qRT-PCR and calculating the interference rate efficiency. Subsequently, the validated siRNA was then employed in a 24-day RNAi experiment. The dry weight of remaining bait and excrement, as well as the wet weight of the sea cucumbers, was measured, enabling calculation of WGR and SGR. Furthermore, the expression levels of three key growth factors (*AjMegf6*, *AjIgf*, and *AjGDF-8*) in the intestinal tissue were analyzed via qPCR.

## Statistical analysis

Statistical analyses were carried out using GraphPad Prism 8.0. Significant differences between groups were determined by one-way analysis of variance (ANOVA) followed by a Tukey's multiple comparisons test. Statistical significance thresholds were defined as follows: $p < 0.05$ (*), $p < 0.01$ (**) and $p < 0.001$ (***), with annotations directly marked on the figures. All data were presented as mean ± SEM from at least three independent biological replicates. Non-linear regression analysis was performed

using a four-parameter logistic equation with automatic outlier exclusion to fit the dose-response curve. The half maximal effective concentration ($EC_{50}$) was derived from the fitted curves through built-in algorithms in GraphPad Prism 8.0.

## Acknowledgements

This research was supported by National Natural Science Foundation of China [grant number 42276103] and Biotechnology and Biological Sciences Research Council (UK; grant numbers BB/M001644/1 and BB/X001024/1).

---

## Additional information

### Funding

| Funder | Grant reference number | Author |
| --- | --- | --- |
| National Natural Science Foundation of China | 42276103 | Muyan Chen |
| Biotechnology and Biological Sciences Research Council | BB/M001644/1 | Maurice R Elphick |
| Biotechnology and Biological Sciences Research Council | BB/X001024/1 | Maurice R Elphick |

The funders had no role in study design, data collection and interpretation, or the decision to submit the work for publication.

### Author contributions

Xiao Cong, Data curation, Software, Formal analysis, Investigation, Visualization, Methodology, Writing – original draft; Huachen Liu, Validation, Investigation, Visualization, Methodology, Writing – review and editing; Lihua Liu, Validation, Methodology; Nayeli Escudero Castelán, Kite GE Jones, Investigation, Visualization, Methodology, Writing – review and editing; Michaela Egertová, Investigation, Visualization, Methodology; Maurice R Elphick, Resources, Formal analysis, Supervision, Funding acquisition, Validation, Visualization, Writing – review and editing; Muyan Chen, Conceptualization, Resources, Formal analysis, Supervision, Funding acquisition, Validation, Visualization, Methodology, Project administration, Writing – review and editing

### Author ORCIDs

Xiao Cong ⓘ https://orcid.org/0009-0001-4325-8015
Huachen Liu ⓘ https://orcid.org/0009-0003-3001-0840
Nayeli Escudero Castelán ⓘ https://orcid.org/0000-0001-8443-9085
Maurice R Elphick ⓘ https://orcid.org/0000-0002-9169-0048
Muyan Chen ⓘ https://orcid.org/0000-0002-3836-1325

### Ethics

All methods were carried out in accordance with the ARRIVE guidelines (https://arriveguidelines.org) for animal experiments. All animal care and use procedures were approved by the Institutional Animal Care and Use Committee of Ocean University of China (Permit Number: 20141201). They were performed according to the Chinese Guidelines for the Care and Use of Laboratory Animals (GB/T 35892-2018).

Reviewer #1 (Public review): https://doi.org/10.7554/eLife.101799.4.sa1
Reviewer #2 (Public review): https://doi.org/10.7554/eLife.101799.4.sa2
Author response https://doi.org/10.7554/eLife.101799.4.sa3

---

## Additional files

### Supplementary files
MDAR checklist

Supplementary file 1. Oligonucleotide sequences used in this study.

### Data availability
All data generated or analyzed during this study are included in the manuscript and supporting files; source data files have been provided for all figures.

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
