## [Editor Report · eLife Assessment]

This **valuable** study characterises receptors for calcitonin-related peptides from a deuterostomian animal, the echinoderm *Apostichopus japonicus*, by a combination of heterologous expression, pharmacological experiments, and the quantification of gene-expression levels. The authors provide **convincing** evidence for a functional calcitonin-related peptide system in the sea cucumber, but further work will be needed to confirm the proposed physiological functions of PDF receptor system in this species. This work should be of interest to scientists studying the signaling pathways, functions, and evolution of neuropeptides, and could be of relevance to improving the culture conditions of this economically key species.

---

## [Referee Report · Reviewer #1 (Public review)]

Summary:

The manuscript characterizes a functional peptidergic system in the echinoderm Apostichopus japonicus that is related to the widely conserved family of calcitonin/diuretic hormone 31 (CT/DH31) peptides in bilaterian animals. In vitro analysis of receptor-ligand interactions, using multiple receptor activation assays, identifies three cognate receptors for two CT-like peptides in the sea cucumber, which stimulate cAMP, calcium, and ERK signaling. Only one of these receptors clusters within the family of calcitonin and calcitonin-like receptors (CTR/CLR) in bilaterian animals, whereas two other receptors cluster with invertebrate pigment dispersing factor receptors (PDFRs). In addition, this study sheds light on the expression and in vivo functions of CT-like peptides in A. japonicus, by quantitative real-time PCR, immunohistochemistry, pharmacological experiments on body wall muscle and intestine preparations, and peptide injection and RNAi knockdown experiments. This reveals a conserved function of CT-like peptides as muscle relaxants and growth regulators in A. japonicus.

Strengths:

This work combines both in vitro and in vivo functional assays to identify a CT-like peptidergic system in an economically relevant echinoderm species, the sea cucumber *A. japonicus*. A major strength of the study is that it identifies three G protein-coupled receptors for AjCT-like peptides, one related to the CTR/CLR family and two related to the PDFR family. A similar finding was previously reported for the CT-related peptide DH31 in *Drosophila melanogaster* that activates both CT-type and PDF-type receptors. Here, the authors expand this observation to a deuterostomian animal, which suggests that receptor promiscuity is a more general feature of the CT/DH31 peptide family and that CT/DH31-like peptides may activate both CT-type and PDF-type receptors in other animals as well.

Besides the identification of receptor-ligand pairs, the downstream signaling pathways of AjCT receptors have been characterized, revealing broad and in some cases receptor-specific effects on cAMP, calcium, and ERK signaling.

Functional characterization of the CT-related peptide system in heterologous cells is complemented with ex vivo and in vivo experiments. First, peptide injection and RNAi knockdown experiments establish transcriptional regulation of all three identified receptors in response to changing AjCT peptide levels. Second, ex vivo experiments reveal a conserved role for the two CT-like peptides as muscle relaxants, which have differential effects on body wall muscle and intestine preparations. Finally, peptide injection and knockdown experiments uncover a growth-promoting role for one CT-like peptide (AjCT2). Injection of AjCT2 at high concentration, or long-term knockdown of the AjCT precursor, affects diverse growth-related parameters including weight gain rate, specific growth rate, and transcript levels of growth-regulating transcription factors. The authors also reveal a growth-promoting function for the PDFR-like receptor AjPDFR2, suggesting that this receptor mediates the effects of AjCT2 on growth.

Weaknesses:

Expression of CT-like peptides was investigated both at transcript and protein level, but insight into the expression of the three peptide receptors is limited. This makes it difficult to understand the mechanism underlying the (different) functions of the two CT-like peptides in vivo. The authors identify differences in signal transduction cascades activated by each peptide, which might underpin distinct functions, but these differences were established only in heterologous cells.

The authors show overlapping phenotypes for a long-term knockdown of the AjCT precursor and the AjPDFR2 receptor, suggesting that the growth-regulating functions of AjCT2 are mediated by this receptor pathway. However, it remains unclear whether this mechanism underpins the growth-regulating function of AjCT2, until further in vivo evidence for this ligand-receptor interaction is presented. For example, the authors could investigate whether knockdown of AjPDFR2 attenuates the effects of AjCT2 peptide injection. In addition, a functional PDF system in this species remains uncharacterized, and a potential role of PDF-like peptides in growth regulation has not yet been investigated in A. japonicus. Therefore, it also remains unclear whether the ability of CT-like peptides to activate PDFRs is an evolutionary ancient property of this peptide family or whether this is an example of convergent evolution in some protostomian (Drosophila) and deuterostomian (sea cucumber) species.

---

## [Referee Report · Reviewer #2 (Public review)]

Summary:

The authors show that A. japonicus calcitonins (AjCT1 and AjCT2) activate not only the calcitonin/calcitonin-like receptor, but they also activate the two "PDF receptors", ex vivo. They also explore secondary messenger pathways that are recruited following receptor activation. They determine the source of CT1 and CT2 using qPCR and in situ hybridization and finally test the effects of these peptides on tissue contractions, feeding and growth. This study provides solid evidence that CT1 and CT2 act as ligands for calcitonin receptors; however, evidence supporting cross-talk between CT peptides and "PDF receptors" is weak.

Strengths:

This is the first study to report pharmacological characterization of CT receptors in an echinoderm. Multiple lines of evidence in cell culture (receptor internalization and secondary messenger pathways) support this conclusion.

---

## [Author Response]

The following is the authors’ response to the previous reviews

**Reviewer #1 (Public review):**
Summary:The manuscript characterizes a functional peptidergic system in the echinoderm Apostichopus japonicus that is related to the widely conserved family of calcitonin/diuretic hormone 31 (CT/DH31) peptides in bilaterian animals. In vitro analysis of receptor-ligand interactions, using multiple receptor activation assays, identifies three cognate receptors for two CT-like peptides in the sea cucumber, which stimulate cAMP, calcium, and ERK signaling. Only one of these receptors clusters within the family of calcitonin and calcitonin-like receptors (CTR/CLR) in bilaterian animals, whereas two other receptors cluster with invertebrate pigment dispersing factor receptors (PDFRs). In addition, this study sheds light on the expression and in vivo functions of CT-like peptides in A. japonicus, by quantitative real-time PCR, immunohistochemistry, pharmacological experiments on body wall muscle and intestine preparations, and peptide injection and RNAi knockdown experiments. This reveals a conserved function of CT-like peptides as muscle relaxants and growth regulators in A. japonicus.Strengths:This work combines both in vitro and in vivo functional assays to identify a CT-like peptidergic system in an economically relevant echinoderm species, the sea cucumber *A. japonicus*. A major strength of the study is that it identifies three G protein-coupled receptors for AjCT-like peptides, one related to the CTR/CLR family and two related to the PDFR family. A similar finding was previously reported for the CT-related peptide DH31 in *Drosophila melanogaster* that activates both CT-type and PDF-type receptors. Here, the authors expand this observation to a deuterostomian animal, which suggests that receptor promiscuity is a more general feature of the CT/DH31 peptide family and that CT/DH31-like peptides may activate both CT-type and PDF-type receptors in other animals as well.Besides the identification of receptor-ligand pairs, the downstream signaling pathways of AjCT receptors have been characterized, revealing broad and in some cases receptor-specific effects on cAMP, calcium, and ERK signaling.Functional characterization of the CT-related peptide system in heterologous cells is complemented with ex vivo and in vivo experiments. First, peptide injection and RNAi knockdown experiments establish transcriptional regulation of all three identified receptors in response to changing AjCT peptide levels. Second, ex vivo experiments reveal a conserved role for the two CT-like peptides as muscle relaxants, which have differential effects on body wall muscle and intestine preparations. Finally, peptide injection and knockdown experiments uncover a growth-promoting role for one CT-like peptide (AjCT2). Injection of AjCT2 at high concentration, or long-term knockdown of the AjCT precursor, affects diverse growth-related parameters including weight gain rate, specific growth rate, and transcript levels of growth-regulating transcription factors. The authors also reveal a growth-promoting function for the PDFR-like receptor AjPDFR2, suggesting that this receptor mediates the effects of AjCT2 on growth.Weaknesses:The authors present a more detailed phylogenetic analysis in the revised version, including a larger number of species. But some clusters in the analysis are not well supported because they have only low bootstrap values. This makes it difficult to interpret the clustering in some parts of the tree.

Thank you for the reviewer’s comments. In response, we have produced a new phylogenetic analysis using the maximum likelihood method. This was done by Nayeli Escudero Castelán and Kite Jones in the Elphick group at QMUL and therefore they have been added as co-authors of this paper. The new phylogenetic tree (Figure 2, line 206) includes broad taxonomic sampling of CT-type receptors and PDF-type receptors. CRH-type receptors, which are also members of the secretin-type GPCR sub-family, have been included as an outgroup to root the tree. In the previous version the much more distantly related vasopressin/oxytocin-type receptors, which are rhodopsin-type GPCRs, were included as an outgroup. Furthermore, VIP-type receptors were also included in the previous tree but these have been omitted from the new tree because VIP receptor orthologs only occur in vertebrates and therefore they are not representative of a bilaterian GPCR family. The new tree shows high bootstrap support for key clades, notably achieving a bootstrap value of 100 for a clade comprising both deuterostomian and protostomian PDF receptors. This provides important evidence that the A. japonicus PDF-type receptors characterised in this study (AjPDFR1, AjPDFR2) are co-orthologs of the PDF-type receptor that has been characterised previously in Drosophila. Similarly, there is strong bootstrap support (100) for a clade comprising CT/DH31-type receptors and, importantly, the CT-type receptor characterised in this study (AjCTR) is positioned in a branch of this clade that comprises deuterostomian CT-type receptors (with bootstrap support of 100). Details of methods employed to produce the new receptor tree are included in lines 727-739. The new phylogenetic tree is shown below and has been incorporated into the revised manuscript (Figure 2, line 206). The description of new phylogenetic tree has also been modified accordingly in the revised manuscript (line 169-183).

References:

Bauknecht P, Jékely G. Large-Scale Combinatorial Deorphanization of Platynereis Neuropeptide GPCRs. Cell reports, 2015, 12(4), 684–693. doi: 10.1016/j.celrep.2015.06.052.

Beets I, Zels S, Vandewyer E, Demeulemeester J, et al. System-wide mapping of peptide-GPCR interactions in *C. elegans*. Cell reports, 2023, 42(9), 113058. doi: 10.1016/j.celrep.2023.113058.

Cardoso J C, Mc Shane J C, Li Z, et al. Revisiting the evolution of Family B1 GPCRs and ligands: Insights from mollusca. Molecular and cellular endocrinology, 2024, 586, 112192. doi: 10.1016/j.mce.2024.112192.

Gorn A H, Lin H Y, Yamin M, et al. Cloning, characterization, and expression of a human calcitonin receptor from an ovarian carcinoma cell line. The Journal of clinical investigation, 1992, 90(5), 1726–1735. doi: 10.1172/JCI116046.

Huang T, Su J, Wang X, et al. Functional Analysis and Tissue-Specific Expression of Calcitonin and CGRP with RAMP-Modulated Receptors CTR and CLR in Chickens. Animals: an open access journal from MDPI, 2024, 14(7), 1058. doi: 10.3390/ani14071058.

Johnson E C, Shafer O T, Trigg J S, et al. A novel diuretic hormone receptor in Drosophila: evidence for conservation of CGRP signaling. Journal of Experimental Biology, 2005, 208(7): 1239-1246. doi: 10.1242/jeb.01529.

McLatchie L M, Fraser N J, Main M J, et al. RAMPs regulate the transport and ligand specificity of the calcitonin-receptor-like receptor. Nature, 1998, 393(6683): 333-339. doi: 10.1038/30666.

Schwartz J, Réalis-Doyelle E, Dubos M P, et al. Characterization of an evolutionarily conserved calcitonin signaling system in a lophotrochozoan, the Pacific oyster (Crassostrea gigas). Journal of Experimental Biology, 2019, 222(13): jeb201319. doi: 10.1242/jeb.201319.

Sekiguchi T, Kuwasako K, Ogasawara M, et al. Evidence for conservation of the calcitonin superfamily and activity-regulating mechanisms in the basal chordate Branchiostoma floridae: insights into the molecular and functional evolution in chordates. Journal of Biological Chemistry, 2016, 291(5): 2345-2356. doi: 10.1074/jbc.M115.664003.

Expression of CT-like peptides was investigated both at transcript and protein level, but insight into the expression of the three peptide receptors is limited. This makes it difficult to understand the mechanism underlying the (different) functions of the two CT-like peptides in vivo. The authors identify differences in signal transduction cascades activated by each peptide, which might underpin distinct functions, but these differences were established only in heterologous cells.

We appreciate the reviewer's insightful comments. Regarding expression of CT-like peptide receptors, we have quantitatively analyzed the mRNA expression levels of the three receptors in key tissues using qRT-PCR (Figure 6, line 319) and receptor expression exhibits significant tissue-specific differences. Combined with the heterologous expression assays and In vivo functional validation, we believe our findings have provided clear mechanistic insights into the functional divergence of the two CT-like peptides. Investigation of the expression of the three receptor proteins in A. japonicus would require generation of specific antibodies, which was beyond the scope of this study. Furthermore, immunohistochemical visualization of neuropeptide receptor expression in other invertebrates has not been reported widely, which likely reflects technical difficulties in generation of antibodies that can be used to specifically detect receptor proteins that are typically expressed a low level in comparison to the neuropeptides that act as their ligands.

We acknowledge that investigating signal transduction cascades in heterologous cells (rather than native A. japonicus cells) is a limitation. However, as a non-model organism, A. japonicus currently lacks established cell lines for such research. Therefore, using heterologous cells was the most feasible approach to examine the differential signaling cascades activated by the peptides through the three receptors. Importantly, our in vivo experiments demonstrated that long-term knockdown of either the AjCT precursor or AjPDFR2 resulted in similar and significant growth defects. The phenotypic consistency strongly suggests that AjCT2 and AjPDFR2 function within the same signaling pathway, with AjPDFR2 serving as the key receptor functionally activated by AjCT2.

The authors show overlapping phenotypes for a long-term knockdown of the AjCT precursor and the AjPDFR2 receptor, suggesting that the growth-regulating functions of AjCT2 are mediated by this receptor pathway. However, it remains unclear whether this mechanism underpins the growth-regulating function of AjCT2, until further in vivo evidence for this ligand-receptor interaction is presented. For example, the authors could investigate whether knockdown of AjPDFR2 attenuates the effects of AjCT2 peptide injection. In addition, a functional PDF system in this species remains uncharacterized, and a potential role of PDF-like peptides in growth regulation has not yet been investigated in A. japonicus. Therefore, it also remains unclear whether the ability of CT-like peptides to activate PDFRs is an evolutionary ancient property of this peptide family or whether this is an example of convergent evolution in some protostomian (Drosophila) and deuterostomian (sea cucumber) species.

Thank you for the reviewer’s insightful comments and constructive questions. We acknowledge the request for more direct evidence to demonstrate how AjCT2 functions in vivo through AjPDFR2. However, long-term knockdown of the AjCT precursor and AjPDFR2 both resulted in identical and significant growth defect phenotypes. The high phenotypic consistency, combined with the activation effect of AjCT2 on AjPDFR2 in heterologous cells, strongly suggests that they function within the same signaling pathway, with AjPDFR2 serving as the key receptor functionally activated by AjCT2. While exogenous peptide injection combined with receptor knockdown is a classic method for verifying receptor activation, phenotypic overlap itself is widely accepted in genetic research as robust evidence for pathway association (Shafer and Taghert, 2009; Van Sinay et al., 2017). A. japonicus is a non-model organism with a 3-month aestivation period in summer followed shortly by winter hibernation. During these periods, we are unable to conduct in vivo experiments. Any single experimental suggestion from reviewers could potentially require one more year of research and we have already conducted an additional year of research, in response to reviewer feedback, since submitting the original manuscript. We hope therefore that these challenges associated with working with aquatic invertebrate non-model organisms is recognized by the reviewers.

We fully agree that the functional PDF/PDFR system in A. japonicus and its potential role in growth regulation remain uncharacterized. Currently, the precursors of the PDF-type neuropeptide in echinoderms remain unidentified, which precludes clear pharmacological characterization of the two receptors. While further exploration of echinoderm PDF-type neuropeptides is still needed, our phylogenetic analysis-conducted using the maximum likelihood method with optimized parameters and rigorous sequence curation-demonstrates that the deuterostomian PDFRs (including AjPDFR1 and AjPDFR2) are positioned in a clade with the well-characterized protostomian PDFR clades with extremely high bootstrap support (value=100). Therefore, these two receptors in A. japonicus clearly belong to the PDF receptor family and our findings clearly indicate that the ability of CT-like peptides to activate PDFRs is either an evolutionarily ancient and conserved property or has arisen independently in different lineages. Details of methods employed to produce the new receptor tree are included in line 727-739. The new phylogenetic tree is shown below and has been incorporated into the revised manuscript (Figure 2, line 206). The description of new phylogenetic tree has also been modified accordingly in the revised manuscript (line 169-183).

References:

Bauknecht P, Jékely G. Large-Scale Combinatorial Deorphanization of Platynereis Neuropeptide GPCRs. Cell reports, 2015, 12(4), 684–693. doi: 10.1016/j.celrep.2015.06.052.

Beets I, Zels S, Vandewyer E, Demeulemeester J, et al. System-wide mapping of peptide-GPCR interactions in *C. elegans*. Cell reports, 2023, 42(9), 113058. doi: 10.1016/j.celrep.2023.113058.

Cardoso J C, Mc Shane J C, Li Z, et al. Revisiting the evolution of Family B1 GPCRs and ligands: Insights from mollusca. Molecular and cellular endocrinology, 2024, 586, 112192. doi: 10.1016/j.mce.2024.112192.

Gorn A H, Lin H Y, Yamin M, et al. Cloning, characterization, and expression of a human calcitonin receptor from an ovarian carcinoma cell line. The Journal of clinical investigation, 1992, 90(5), 1726–1735. doi: 10.1172/JCI116046.

Huang T, Su J, Wang X, et al. Functional Analysis and Tissue-Specific Expression of Calcitonin and CGRP with RAMP-Modulated Receptors CTR and CLR in Chickens. Animals: an open access journal from MDPI, 2024, 14(7), 1058. doi: 10.3390/ani14071058.

Johnson E C, Shafer O T, Trigg J S, et al. A novel diuretic hormone receptor in Drosophila: evidence for conservation of CGRP signaling. Journal of Experimental Biology, 2005, 208(7): 1239-1246. doi: 10.1242/jeb.01529.

McLatchie L M, Fraser N J, Main M J, et al. RAMPs regulate the transport and ligand specificity of the calcitonin-receptor-like receptor. Nature, 1998, 393(6683): 333-339. doi: 10.1038/30666.

Schwartz J, Réalis-Doyelle E, Dubos M P, et al. Characterization of an evolutionarily conserved calcitonin signaling system in a lophotrochozoan, the Pacific oyster (Crassostrea gigas). Journal of Experimental Biology, 2019, 222(13): jeb201319. doi: 10.1242/jeb.201319.

Sekiguchi T, Kuwasako K, Ogasawara M, et al. Evidence for conservation of the calcitonin superfamily and activity-regulating mechanisms in the basal chordate Branchiostoma floridae: insights into the molecular and functional evolution in chordates. Journal of Biological Chemistry, 2016, 291(5): 2345-2356. doi: 10.1074/jbc.M115.664003.

Shafer, O. T., & Taghert, P. H. (2009). RNA-interference knockdown of Drosophila pigment dispersing factor in neuronal subsets: the anatomical basis of a neuropeptide's circadian functions. PloS one, 4(12), e8298. doi: 10.1371/journal.pone.0008298.

Van Sinay, E., Mirabeau, O., Depuydt, G., Van Hiel, M. B., Peymen, K., Watteyne, J., Zels, S., Schoofs, L., & Beets, I. (2017). Evolutionarily conserved TRH neuropeptide pathway regulates growth in *Caenorhabditis elegans*. Proceedings of the National Academy of Sciences of the United States of America, 114(20), E4065–E4074. doi: 10.1073/pnas.1617392114.

**Reviewer #2 (Public review):**
Summary:The authors show that A. japonicus calcitonins (AjCT1 and AjCT2) activate not only the calcitonin/calcitonin-like receptor, but they also activate the two "PDF receptors", ex vivo. They also explore secondary messenger pathways that are recruited following receptor activation. They determine the source of CT1 and CT2 using qPCR and in situ hybridization and finally test the effects of these peptides on tissue contractions, feeding and growth. This study provides solid evidence that CT1 and CT2 act as ligands for calcitonin receptors; however, evidence supporting cross-talk between CT peptides and "PDF receptors" is weak.Strengths:This is the first study to report pharmacological characterization of CT receptors in an echinoderm. Multiple lines of evidence in cell culture (receptor internalization and secondary messenger pathways) support this conclusion.Weaknesses:The authors claim that A. japonicus CTs activate "PDF" receptors and suggest that this cross-talk is evolutionary ancient since similar phenomenon also exists in the fly *Drosophila melanogaster*. These conclusions are not fully supported. The authors perform phylogenetic analysis to show that the two "PDF" receptors form an independent clade. The bootstrap support is quite low in a lot of instances, especially for the deuterostomian and protostomian PDFR clades which is below 30. With such low support, it is unclear if the clade comprising deuterostomian "PDFR" is in fact PDFRs and not another receptor type whose endogenous ligand (besides CT) remains to be discovered.

Thank you for the reviewer’s comments. In response, we have produced a new phylogenetic analysis using the maximum likelihood method. This was done by Nayeli Escudero Castelán and Kite Jones in the Elphick group at QMUL and therefore they have been added as co-authors of this paper. The new phylogenetic tree (Figure 2, line 206) includes broad taxonomic sampling of CT-type receptors and PDF-type receptors. CRH-type receptors, which are also members of the secretin-type GPCR sub-family, have been included as an outgroup to root the tree. In the previous version the much more distantly related vasopressin/oxytocin-type receptors, which are rhodopsin-type GPCRs, were included as an outgroup. Furthermore, VIP-type receptors were also included in the previous tree but these have been omitted from the new tree because VIP receptor orthologs only occur in vertebrates and therefore they are not representative of a bilaterian GPCR family. The new tree shows high bootstrap support for key clades, notably achieving a bootstrap value of 100 for a clade comprising both deuterostomian and protostomian PDF receptors. This provides important evidence that the A. japonicus PDF-type receptors characterized in this study (AjPDFR1, AjPDFR2) are co-orthologs of the PDF-type receptor that has been characterized previously in Drosophila. Similarly, there is strong bootstrap support (100) for a clade comprising CT/DH31-type receptors and, importantly, the CT-type receptor characterized in this study (AjCTR) is positioned in a branch of this clade that comprises deuterostomian CT-type receptors (with bootstrap support of 100). Details of methods employed to produce the new receptor tree are included in lines 727-739. The new phylogenetic tree is shown below and has been incorporated into the revised manuscript (Figure 2, line 206). The description of new phylogenetic tree has also been modified accordingly in the revised manuscript (line 169-183).

References:

Bauknecht P, Jékely G. Large-Scale Combinatorial Deorphanization of Platynereis Neuropeptide GPCRs. Cell reports, 2015, 12(4), 684–693. doi: 10.1016/j.celrep.2015.06.052.

Beets I, Zels S, Vandewyer E, Demeulemeester J, et al. System-wide mapping of peptide-GPCR interactions in *C. elegans*. Cell reports, 2023, 42(9), 113058. doi: 10.1016/j.celrep.2023.113058.

Cardoso J C, Mc Shane J C, Li Z, et al. Revisiting the evolution of Family B1 GPCRs and ligands: Insights from mollusca. Molecular and cellular endocrinology, 2024, 586, 112192. doi: 10.1016/j.mce.2024.112192.

Gorn A H, Lin H Y, Yamin M, et al. Cloning, characterization, and expression of a human calcitonin receptor from an ovarian carcinoma cell line. The Journal of clinical investigation, 1992, 90(5), 1726–1735. doi: 10.1172/JCI116046.

Huang T, Su J, Wang X, et al. Functional Analysis and Tissue-Specific Expression of Calcitonin and CGRP with RAMP-Modulated Receptors CTR and CLR in Chickens. Animals: an open access journal from MDPI, 2024, 14(7), 1058. doi: 10.3390/ani14071058.

Johnson E C, Shafer O T, Trigg J S, et al. A novel diuretic hormone receptor in Drosophila: evidence for conservation of CGRP signaling. Journal of Experimental Biology, 2005, 208(7): 1239-1246. doi: 10.1242/jeb.01529.

McLatchie L M, Fraser N J, Main M J, et al. RAMPs regulate the transport and ligand specificity of the calcitonin-receptor-like receptor. Nature, 1998, 393(6683): 333-339. doi: 10.1038/30666.

Schwartz J, Réalis-Doyelle E, Dubos M P, et al. Characterization of an evolutionarily conserved calcitonin signaling system in a lophotrochozoan, the Pacific oyster (Crassostrea gigas). Journal of Experimental Biology, 2019, 222(13): jeb201319. doi: 10.1242/jeb.201319.

Sekiguchi T, Kuwasako K, Ogasawara M, et al. Evidence for conservation of the calcitonin superfamily and activity-regulating mechanisms in the basal chordate Branchiostoma floridae: insights into the molecular and functional evolution in chordates. Journal of Biological Chemistry, 2016, 291(5): 2345-2356. doi: 10.1074/jbc.M115.664003.

**Reviewer #2 (Recommendations for the authors):**
Figure 1C: The bootstrap support is quite low in a lot of instances, especially for the deuterostomian and protostomian PDFR clades which is below 30. With such support, I would be hesitant to label the blue clade as deuterostomian PDFR for two reasons: (1) no members of this clade have been shown to be activated by a PDF-like substance and (2) the current study shows that these receptors are activated by CT-type peptides. Therefore, the phylogenetic analyses do not support the conclusions of this paper. What is the basis for calling these receptors PDFR and not CTR in light of weak phylogenetic support?

Thank you for the reviewer’s comments. In response, we have produced a new phylogenetic analysis using the maximum likelihood method. This was done by Nayeli Escudero Castelán and Kite Jones in the Elphick group at QMUL and therefore they have been added as co-authors of this paper. The new phylogenetic tree (Figure 2, line 206) includes broad taxonomic sampling of CT-type receptors and PDF-type receptors. CRH-type receptors, which are also members of the secretin-type GPCR sub-family, have been included as an outgroup to root the tree. In the previous version the much more distantly related vasopressin/oxytocin-type receptors, which are rhodopsin-type GPCRs, were included as an outgroup. Furthermore, VIP-type receptors were also included in the previous tree but these have been omitted from the new tree because VIP receptor orthologs only occur in vertebrates and therefore they are not representative of a bilaterian GPCR family. The new tree shows high bootstrap support for key clades, notably achieving a bootstrap value of 100 for a clade comprising both deuterostomian and protostomian PDF receptors. This provides important evidence that the A. japonicus PDF-type receptors characterized in this study (AjPDFR1, AjPDFR2) are co-orthologs of the PDF-type receptor that has been characterized previously in Drosophila. Similarly, there is strong bootstrap support (100) for a clade comprising CT/DH31-type receptors and, importantly, the CT-type receptor characterized in this study (AjCTR) is positioned in a branch of this clade that comprises deuterostomian CT-type receptors (with bootstrap support of 100). Details of methods employed to produce the new receptor tree are included in lines 727-739 The new phylogenetic tree is shown below and has been incorporated into the revised manuscript (Figure 2, line 206). The description of new phylogenetic tree has also been modified accordingly in the revised manuscript (line 169-183).

We agree with the reviewer that no members of the PDF-type receptor clade in deuterostomes have yet been shown to be activated by a PDF-like substance. That is because the precursors of the PDF-type neuropeptides in echinoderms remain unidentified so far, which precludes clear pharmacological characterization of these receptors within the deuterostomian PDFR clade. However, the new phylogenetic tree now provides strong support (bootstrap value = 100) for the clade comprising deuterostomian and protostomian PDFRs, confirming the classification of AjPDFR1 and AjPDFR2 as PDF-type receptors.

References:

Bauknecht P, Jékely G. Large-Scale Combinatorial Deorphanization of Platynereis Neuropeptide GPCRs. Cell reports, 2015, 12(4), 684–693. doi: 10.1016/j.celrep.2015.06.052.

Beets I, Zels S, Vandewyer E, Demeulemeester J, et al. System-wide mapping of peptide-GPCR interactions in *C. elegans*. Cell reports, 2023, 42(9), 113058. doi: 10.1016/j.celrep.2023.113058.

Cardoso J C, Mc Shane J C, Li Z, et al. Revisiting the evolution of Family B1 GPCRs and ligands: Insights from mollusca. Molecular and cellular endocrinology, 2024, 586, 112192. doi: 10.1016/j.mce.2024.112192.

Gorn A H, Lin H Y, Yamin M, et al. Cloning, characterization, and expression of a human calcitonin receptor from an ovarian carcinoma cell line. The Journal of clinical investigation, 1992, 90(5), 1726–1735. doi: 10.1172/JCI116046.

Huang T, Su J, Wang X, et al. Functional Analysis and Tissue-Specific Expression of Calcitonin and CGRP with RAMP-Modulated Receptors CTR and CLR in Chickens. Animals: an open access journal from MDPI, 2024, 14(7), 1058. doi: 10.3390/ani14071058.

Johnson E C, Shafer O T, Trigg J S, et al. A novel diuretic hormone receptor in Drosophila: evidence for conservation of CGRP signaling. Journal of Experimental Biology, 2005, 208(7): 1239-1246. doi: 10.1242/jeb.01529.

McLatchie L M, Fraser N J, Main M J, et al. RAMPs regulate the transport and ligand specificity of the calcitonin-receptor-like receptor. Nature, 1998, 393(6683): 333-339. doi: 10.1038/30666.

Schwartz J, Réalis-Doyelle E, Dubos M P, et al. Characterization of an evolutionarily conserved calcitonin signaling system in a lophotrochozoan, the Pacific oyster (Crassostrea gigas). Journal of Experimental Biology, 2019, 222(13): jeb201319. doi: 10.1242/jeb.201319.

Sekiguchi T, Kuwasako K, Ogasawara M, et al. Evidence for conservation of the calcitonin superfamily and activity-regulating mechanisms in the basal chordate Branchiostoma floridae: insights into the molecular and functional evolution in chordates. Journal of Biological Chemistry, 2016, 291(5): 2345-2356. doi: 10.1074/jbc.M115.664003.

The new results following AjCT and AjPDFR2 knockdown are a welcome addition. While this additional evidence supports the claim that AjCT could mediate its effects via AjPDFR2, this evidence does not show that AjCT acts as an endogenous ligand for PDFR in vivo. In combination with the weak phylogenetic analyses, I would recommend the authors to key down their claims that they have functionally characterized a PDFR (in the title and text).

Thank you for your insightful comments and we do understand the reviewer’s concern.

Regarding “the weak phylogenetic analyses”, as highlighted above, we have produced a new phylogenetic tree (Fig 2, line 206) that provides strong bootstrap support for the clade comprising deuterostome and protostome PDF-type receptors. For this reason, it is our opinion that inclusion of “pigment-dispersing factor-type receptors” in the title of the paper is appropriate. The details of phylogenetic analysis method were added in line 727-739, and the updated phylogenetic tree has been incorporated into the revised manuscript (Figure 2, line 206). The description of new phylogenetic tree has also been modified accordingly in the revised manuscript (line 169-183). Besides, long-term knockdown of the AjCT precursor and AjPDFR2 both resulted in identical and significant growth defect phenotypes. And the observation of phenotypic overlap is widely accepted in genetic research as strong evidence for pathway association (Shafer and Taghert, 2009; Van Sinay et al., 2017). This high degree of phenotypic consistency, coupled with our in vitro finding that AjCT2 specifically activates AjPDFR2, strongly supports the conclusion that AjCT2 and AjPDFR2 function within the same signaling pathway in vivo, with AjPDFR2 serving as the key receptor functionally activated by AjCT2.

References:

Shafer, O. T., & Taghert, P. H. (2009). RNA-interference knockdown of Drosophila pigment dispersing factor in neuronal subsets: the anatomical basis of a neuropeptide's circadian functions. PloS one, 4(12), e8298. doi: 10.1371/journal.pone.0008298.

Van Sinay, E., Mirabeau, O., Depuydt, G., Van Hiel, M. B., Peymen, K., Watteyne, J., Zels, S., Schoofs, L., & Beets, I. (2017). Evolutionarily conserved TRH neuropeptide pathway regulates growth in *Caenorhabditis elegans*. Proceedings of the National Academy of Sciences of the United States of America, 114(20), E4065–E4074. doi: 10.1073/pnas.1617392114.

Since there is no formal logic defining the use of "type" vs "like" vs "related", I would encourage the authors to use one term (of their choice) to avoid unnecessary confusion. Or another possibility is that these relationships are defined at some point in the manuscript so that it becomes clear to the reader.

Thank you for the reviewer’s comments. The “CT-related peptides” has defined in the Introduction (line 54-58). As per your suggestion, we have now defined both “CT-type peptides” and “CT-like peptides” in the Introduction (line 76-79). “CT-type peptides” are characterized by an N-terminal disulphide bridge, whereas “CT-like peptides” (diuretic hormone 31 (DH31)-type peptides) lack this feature. Additionally, in accordance with the definitions, we have corrected these three descriptions in the revised manuscript (line 80, 83, 88 for “CT-type peptides”) to ensure consistent and accurate usage of these terms.

"To provide in vivo evidence supporting CT-mediated activation of "PDF" receptors, we conducted the following experiments: Firstly, we confirmed that AjPDFR1 and AjPDFR2were the functional receptors of AjCT1and AjCT2 (Figure 2, 3 and 4). Secondly, injection of AjCT2 and siAjCTP1/2-1 in vivo induced corresponding changes in AjPDFR1and AjPDFR2expression levels in the intestine (Figure 8C, 9A, 9B and 9C)."None of these experiments provide direct evidence that CT activates PDFR in vivo. The functional studies are indeed a welcome addition but they cannot discriminate between correlation and causation.

Thank you for the reviewer’s insightful comments. We agree that the functional studies do not constitute direct proof that CT’s activation of PDFR in vivo. However, we observed identical and significant growth defect phenotypes following long-term knockdown of the AjCT precursor and the AjPDFR2. This high degree of phenotypic congruence, combined with the established in vitro activation of AjPDFR2 by AjCT2, provides strong support for the conclusion that AjCT2 acts as the key endogenous ligand activating the AjPDFR2 signaling pathway in vivo. Importantly, such phenotypic overlap has been widely accepted in genetic research as strong evidence for functional pathway association (Shafer and Taghert, 2009; Van Sinay et al., 2017).

References:

Shafer, O. T., & Taghert, P. H. (2009). RNA-interference knockdown of Drosophila pigment dispersing factor in neuronal subsets: the anatomical basis of a neuropeptide's circadian functions. PloS one, 4(12), e8298. doi: 10.1371/journal.pone.0008298.

Van Sinay, E., Mirabeau, O., Depuydt, G., Van Hiel, M. B., Peymen, K., Watteyne, J., Zels, S., Schoofs, L., & Beets, I. (2017). Evolutionarily conserved TRH neuropeptide pathway regulates growth in *Caenorhabditis elegans*. Proceedings of the National Academy of Sciences of the United States of America, 114(20), E4065–E4074. doi: 10.1073/pnas.1617392114.